# The SWI/SNF chromatin remodeling assemblies BAF and PBAF differentially regulate cell cycle exit and cellular invasion *in vivo*

**Jayson J. Smith**[1], **Yutong Xiao**[1], **Nithin Parsan**[1,2], **Taylor N. Medwig-Kinney**[1], **Michael A. Q. Martinez**[1], **Frances E. Q. Moore**[1], **Nicholas J. Palmisano**[1], **Abraham Q. Kohrman**[1,3], **Mana Chandhok Delos Reyes**[1], **Rebecca C. Adikes**[1,4], **Simeiyun Liu**[1,5], **Sydney A. Bracht**[1,6], **Wan Zhang**[1], **Kailong Wen**[7,8], **Paschalis Kratsios**[7,8], **David Q. Matus**[1‡*]

1 Department of Biochemistry and Cell Biology, Stony Brook University, Stony Brook, New York, United States of America, 2 Massachusetts Institute of Technology, Cambridge, Massachusetts, United States of America, 3 Department of Molecular Biology, Princeton University, Princeton, New Jersey, United States of America, 4 Biology Department, Siena College, Loudonville, New York, United States of America, 5 Molecular, Cellular and Developmental Biology, University of California Santa Cruz, Santa Cruz, California, United States of America, 6 Department of Cell Biology, Johns Hopkins University, Baltimore, Maryland, United States of America, 7 The Grossman Institute for Neuroscience, Quantitative Biology, and Human Behavior, University of Chicago, Chicago, Illinois, United States of America, 8 Department of Neurobiology, University of Chicago, Chicago, Illinois, United States of America

‡ Lead Contact.
* david.matus@stonybrook.edu

**Data Availability Statement:** All relevant data are within the manuscript and its Supporting Information files.

## Abstract

Chromatin remodelers such as the SWI/SNF complex coordinate metazoan development through broad regulation of chromatin accessibility and transcription, ensuring normal cell cycle control and cellular differentiation in a lineage-specific and temporally restricted manner. Mutations in genes encoding the structural subunits of chromatin, such as histone subunits, and chromatin regulating factors are associated with a variety of disease mechanisms including cancer metastasis, in which cancer co-opts cellular invasion programs functioning in healthy cells during development. Here we utilize *Caenorhabditis elegans* anchor cell (AC) invasion as an *in vivo* model to identify the suite of chromatin agents and chromatin regulating factors that promote cellular invasiveness. We demonstrate that the SWI/SNF ATP-dependent chromatin remodeling complex is a critical regulator of AC invasion, with pleiotropic effects on both $G_0$ cell cycle arrest and activation of invasive machinery. Using targeted protein degradation and enhanced RNA interference (RNAi) vectors, we show that SWI/SNF contributes to AC invasion in a dose-dependent fashion, with lower levels of activity in the AC corresponding to aberrant cell cycle entry and increased loss of invasion. Our data specifically implicate the SWI/SNF BAF assembly in the regulation of the $G_0$ cell cycle arrest in the AC, whereas the SWI/SNF PBAF assembly promotes AC invasion via cell cycle-independent mechanisms, including attachment to the basement membrane (BM) and activation of the pro-invasive *fos-1*/FOS gene. Together these findings demonstrate that the SWI/SNF complex is necessary for two essential components of AC invasion:

**Funding:** D.Q.M. is a Damon Runyon-Rachleff Innovator supported by the Damon Runyon Cancer Research Foundation [DRR-47-17]. This work is also supported by research grants to D.Q.M. from the National Institute of General Medical Sciences (NIGMS) [R01GM121597]. J.J.S. received support from the W. Burghardt Turner Fellowship. J.J.S. [3R01GM121597-02S1], MA.Q.M [3R01GM121597-03S1] and F.E.Q.M. [3R01GM121597-04S1] were supported through NIGMS Diversity Supplements to R01GM121597. M.A.Q.M. was also supported by the National Cancer Institute (NCI) 1F30CA257383-01A1. A.Q.K. was supported by NIGMS [F31GM128319]. R.C.A. was supported by NIGMS [F32GM1283190]. T.N. M-K. was supported by NICHD [F31HD100091]. N. J.P. was supported by the American Cancer Society [132969-PF-18-226-01-CSM]. P.K. was supported by the National Institute of Neurological Disorders and Stroke [R01NS118078]. The funders had no role in study design, data collection and analysis, decision to publish, or preparation of the manuscript.

**Competing interests:** The authors have declared that no competing interests exist.

arresting cell cycle progression and remodeling the BM. The work here provides valuable single-cell mechanistic insight into how the SWI/SNF assemblies differentially contribute to cellular invasion and how SWI/SNF subunit-specific disruptions may contribute to tumorigeneses and cancer metastasis.

## Author summary

Cellular invasion is required for animal development and homeostasis. Inappropriate activation of invasion however can result in cancer metastasis. Invasion programs are orchestrated by complex gene regulatory networks (GRN) that function in a coordinated fashion to turn on and off pro-invasive genes. While the core of GRNs are DNA binding transcription factors, they require aid from chromatin remodelers to access the genome. To identify the suite of pro-invasive chromatin remodelers, we paired high resolution imaging with RNA interference to individually knockdown 269 chromatin factors, identifying the evolutionarily conserved SWItching defective/Sucrose Non-Fermenting (SWI/SNF) ATP-dependent chromatin remodeling complex as a new regulator of *Caenorhabditis elegans* anchor cell (AC) invasion. Using a combination of CRISPR/Cas9 genome engineering and targeted protein degradation we demonstrate that the core SWI/SNF complex functions in a dose-dependent manner to control invasion. Further, we determine that the accessory SWI/SNF complexes, BAF and PBAF, contribute to invasion via distinctive mechanisms: BAF is required to prevent inappropriate proliferation while PBAF promotes AC attachment and remodeling of the basement membrane. Together, our data provide insights into how the SWI/SNF complex, which is mutated in many human cancers, can function in a dose-dependent fashion to regulate switching from invasive to proliferative fates.

## Introduction

Cellular invasion through basement membranes (BMs) is a critical step in metazoan development and is important for human health and fitness. Early in hominid development, trophoblasts must invade into the maternal endometrium for proper blastocyst implantation [1]. In the context of immunity, leukocytes become invasive upon injury or infection to travel between the bloodstream and interstitial tissues [2, 3]. Atypical activation of invasive behavior is associated with a variety of diseases, including rheumatoid arthritis wherein fibroblast-like synoviocytes adopt invasive cellular behavior, leading to the expansion of arthritic damage to previously unaffected joints [4, 5]. Aberrant activation of cell invasion is also one of the hallmarks of cancer metastasis [6].

A variety of *in vitro* and *in vivo* models have been developed to study the process of cellular invasion at the genetic and cellular levels. *In vitro* invasion assays typically involve 3D hydrogel lattices, such as Matrigel, through which cultured metastatic cancer cells will invade in response to chemo-attractants [7]. Recently, microfluidic systems have been integrated with collagen matrices to improve these *in vitro* investigations of cellular invasion [8]. While *in vitro* invasion models provide an efficient means to study the mechanical aspects of cellular invasion, they are currently unable to replicate the complex microenvironment in which cells must invade during animal development and disease. A variety of *in vivo* invasion models have been studied, including cancer xenograft models in mouse [9–11] and zebrafish [12, 13],

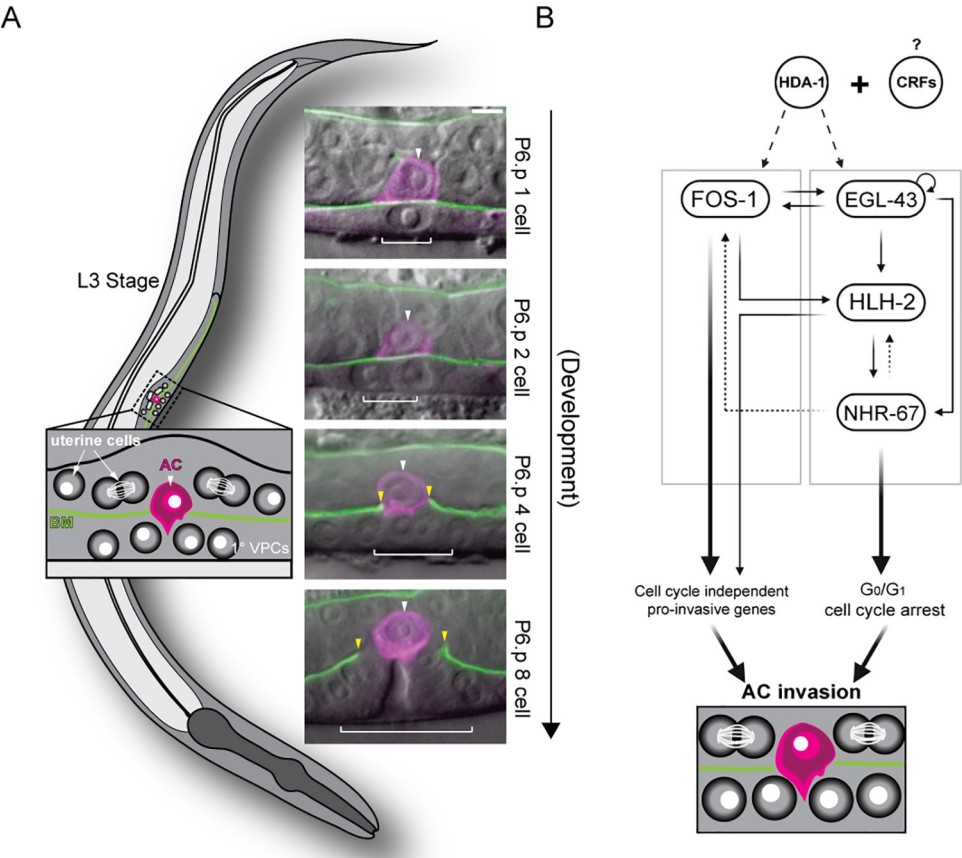

**Fig 1. Summary of *C. elegans* AC invasion through the underlying BM. (A)** Schematic depicting AC invasion in the mid-L3 stage of *C. elegans* development (left) and micrographs demonstrating the coordination of AC (magenta, *cdh-3p::PH::mCherry*) invasion through the BM (green, *laminin::GFP*) with primary vulval development in the uterine-specific RNAi hypersensitive background used in the chromatin factor RNAi screen. The fluorescent AC-specific membrane marker and BM marker are overlaid on DIC in each image. White arrowheads indicate ACs, yellow arrowheads indicate boundaries of breach in the BM, and white brackets indicate 1˚ VPCs. Scale bar, 5 μm. **(B)** Overview of the transcription factor GRN governing AC invasion [22, 24], which consists of cell cycle-independent (*fos-1*) and dependent (*egl-43*, *hlh-2*, and *nhr-67*) subcircuits, which together with *hda-1* promote pro-invasive gene expression and maintain cell cycle arrest in the AC.

each having their respective benefits and drawbacks. Over the past ~15 years, *Caenorhabditis elegans* anchor cell (AC) invasion has emerged as a powerful alternative model due to its visually tractable single-cell nature (**Fig 1A**) [14].

Previous work demonstrated a high degree of evolutionary conservation in the cell-autonomous mechanisms underlying BM invasion [3, 15], including basolateral polarization of the F-actin cytoskeleton/cytoskeletal regulators and the expression of matrix metalloproteinases (MMPs) [16–21]. Moreover, in order to breach the BM, the AC requires the expression of transcription factors (TFs), whose human homologs are common to metastatic cancers, including *egl-43* (EVI1/MEL), *fos-1* (FOS), *hlh-2* (E/Daughterless), and *nhr-67* (TLX/Tailless) [22] (**Fig 1B**). In addition to the expression of pro-invasive genes, there is increasing evidence that cells must also arrest in the cell cycle to adopt an invasive phenotype [23]. Our previous work has demonstrated that the AC must terminally differentiate and arrest in the $G_0/G_1$ phase of the cell cycle to invade the BM and make contact with the underlying primary vulval

precursor cells (1° VPCs) [22, 24]. The regulatory mechanisms that couple $G_0/G_1$ cell cycle arrest with the ability of a cell to invade the BM remain unclear.

Cell-extrinsic and cell-intrinsic factors, such as chromatin remodeling complexes and TFs, control many aspects of cell fate from plasticity to terminal differentiation and cell cycle arrest. This decision between plasticity and specification is in part the consequence of a complex, genome-wide antagonism between Polycomb group (PcG) transcriptional repression and Trithorax group (TrxG) transcriptional activation [25–27]. One example of this is the binding of pioneer TFs OCT4 and SOX2 to target DNA in order to retain pluripotency in murine embryonic stem cells; the association of these TFs with their targets has been characterized as an indirect consequence of chromatin accessibility at these target regions [28]. A recent study has shown that chromatin accessibility of enhancers in crucial cell cycle genes which promote the $G_1/S$ transition, including Cyclin E and E2F transcription factor 1, is developmentally restricted to reinforce terminal differentiation and cell cycle exit during *Drosophila melanogaster* pupal wing morphogenesis [29]. In *C. elegans* myogenesis, the SWItching defective/ Sucrose Non-Fermenting (SWI/SNF) ATP-dependent chromatin remodeling complex, a member of the TrxG family of complexes, both regulates the expression of the MyoD transcription factor (*hlh-1*) and acts redundantly to promote differentiation and $G_0$ cell cycle arrest with several core cell cycle regulators including cullin 1 (CUL1/*cul-1)*, cyclin-dependent kinase inhibitor 1 (*cki-1)*, FZR1 (*fzr*-1), and the RB transcriptional corepressor (RBL1/*lin*-35) [30]. The importance of the dynamic regulation of chromatin states for the acquisition and implementation of differentiated behaviors is also reflected in the *C. elegans* AC, as previous work has shown that the histone deacetylase *hda-1* (HDAC1/2) is required for pro-invasive gene expression and therefore the differentiated behavior of cellular invasion [24] (**Fig 1B**).

A comprehensive investigation of the regulatory mechanism(s) governing AC invasion should include a thorough description of the suite of chromatin agents and chromatin regulating factors that are required for $G_0/G_1$ cell cycle arrest and invasive differentiation in the AC. In this study we perform an RNA interference (RNAi) screen in *C. elegans*, specifically focusing on genes involved in chromatin structure and remodeling or histone modification (collectively called "chromatin factors"). We identify 82 chromatin factors whose transcriptional depletion resulted in significant AC invasion defects. Among the 82 hits recovered in the screen, the SWI/SNF complex emerged as the most well-represented single chromatin remodeling complex. RNAi knockdown of subunits specific to the SWI/SNF core (*swsn-1* and *snfc-5/ swsn-5*), and both BAF (BRG/BRM-Associated Factors; *swsn-8/let-526*) and PBAF (Polybromo Associated BAF; *pbrm-1* and *swsn-7*) assemblies resulted in penetrant loss of AC invasion. We generated fluorescent reporter knock-in alleles of subunits of the core (*GFP::swsn-4*) and BAF (*swsn-8::GFP*) assembly of the SWI/SNF complex using CRISPR/Cas9-mediated genome engineering. These alleles, used in conjunction with an endogenously labeled PBAF (*pbrm-1:: eGFP*) assembly subunit, enabled us to determine the developmental dynamics of the SWI/ SNF ATPase and assembly-specific subunits, gauge the efficiency of various SWI/SNF knockdown strategies, and assess intra-complex and inter-assembly regulation. Using improved RNAi constructs and an anti-GFP nanobody degradation strategy [31], we demonstrated that the cell autonomous contribution of the SWI/SNF complex to AC invasion is dose dependent. This finding parallels similar studies in cancer [32–35] and *C. elegans* mesoblast development [36]. Surprisingly, examination using a CDK activity sensor [37] revealed assembly-specific contributions to AC invasion: whereas BAF promotes AC invasion in a cell cycle-dependent manner, PBAF contributes to invasion in a cell cycle-independent manner. Finally, we utilized the auxin-inducible degron (AID) system combined with PBAF RNAi to achieve strong combinatorial PBAF subunit depletion in the AC, which resulted in loss of both AC invasion and adhesion to the BM. Together, these findings provide insight into how the SWI/SNF complex

assemblies may contribute to distinct aspects of proliferation and metastasis in human malignancies.

## Results

### An RNAi screen of 269 chromatin factors identifies SWI/SNF as a key regulator of AC invasion

To identify the suite of chromatin factors that, along with *hda-1*, contribute to AC invasion, we generated an RNAi sub-library of 269 RNAi clones from the complete Vidal RNAi library and a subset of the Ahringer RNAi library [38, 39] targeting genes implicated in chromatin state, chromatin remodeling, or histone modification (**Fig 1B** and **S1 Table**). Because chromatin regulatory factors act globally to control gene expression, we screened each RNAi clone by high-resolution differential interference contrast (DIC) and epifluorescence microscopy in a uterine-specific RNAi hypersensitive background containing labeled BM (*laminin*::*GFP*) and an AC reporter (*cdh-3p*::*PH*::*mCherry*) (**Fig 1A** and **S1 Table**) [14, 22, 24, 40]. This genetic background allowed us to limit the effect of RNAi transcriptional knockdown of chromatin factors to the AC and the neighboring uterine tissue, and only for a time period following the specification of the AC [40]. As the neighboring uterine cells do not contribute to the invasion program [14], AC invasion defects following RNAi treatments in this background are indicative of cell autonomous pro-invasive gene function [24, 40]. In wild-type animals, by the time the 1˚ fated P6.p vulval precursor cell has divided twice (P6.p 4-cell stage), 100% of ACs have successfully breached the underlying BMs and made contact with the P6.p grand-daughters [14]. Similarly, we found that all ACs invaded in the uterine-specific RNAi hypersensitive strain used in our RNAi screen, though we observed a low penetrance of ACs with a delay in the timing of invasion, such that at the P6.p 4-cell stage, when we scored invasion, 2% (2/100 animals) still had an intact BM. Thus, we used this baseline defect as a statistical reference point for this genetic background. We defined the cut-off threshold for significant defects in invasion following RNAi treatment as those RNAi clones that resulted in loss of invasion in at least ~13% of treated animals (4/30 animals, Fisher's exact test = 0.0252). By this threshold, we recovered 82 chromatin factors (30.5% of total screened) that significantly regulate AC invasion (**S2 Table**). The finding that loss of nearly a third of the chromatin factors included in the RNAi screen results in significant AC invasion defects suggests a general requirement for regulation of chromatin states in the acquisition of invasive behavior. Interestingly, five subunits of the broadly conserved SWI/SNF chromatin remodeling complex were recovered as significant regulators of AC invasion: *swsn-1*(SMARCC1/SMARCC2; 23% AC invasion defect), *swsn-5/snfc-5* (SMARCB1; 20% AC invasion defect), *swsn-7* (ARID2; 23% AC invasion defect), *swsn-8/let-526* (ARID1A/ARID1B; 23% AC invasion defect), and *pbrm-1* (PBRM1; 20% AC invasion defect) (**S2 Table**). As such, SWI/SNF is well-represented among the roster of significant regulators of AC invasion identified in the screen, with representation of the core (*swsn-1* and *swsn-5*), BAF (*swsn-8*) and PBAF (*pbrm-1* and *swsn-7*) assemblies. Given the prevalence of SWI/SNF subunits recovered as significant regulators of AC invasion in our RNAi screen and the crucial role SWI/SNF plays in the regulation of animal development [41–46], tumorigenesis [33, 47–49], and cell cycle control [30, 36, 50–52], we chose to focus our efforts on characterizing the role of the SWI/SNF complex in promoting AC invasion.

To confirm our RNAi results implicating the SWI/SNF complex as an activator of AC invasion, we obtained two temperature sensitive hypomorphic alleles, *swsn-1(os22)* and *swsn-4 (os13)* [42], and scored for defects in AC invasion in a genetic background containing both BM (*laminin*::*GFP*) and AC (*cdh-3p*::*mCherry*::*moeABD*) reporters. While we observed no defects in AC invasion in animals grown at the permissive temperature (15˚C) (**S1A Fig**),

animals containing hypomorphic alleles for core subunits *swsn-1* and *swsn-4* cultured at the restrictive temperature (25˚C) displayed defects in 20% (10/50) and 24% (12/50) of animals, respectively (**S1B Fig**). These data with the *swsn-1(os22)* allele corroborated our *swsn-1(RNAi)* data from the chromatin factor RNAi screen. Additionally, since neither of the RNAi libraries used to compose the chromatin factor screen in this study (see above) contained a *swsn-4 (RNAi)* clone, results with the *swsn-4(os13)* allele also complement data from our RNAi screen by suggesting that AC invasion depends on the expression of the sole *C. elegans* SWI/SNF ATPase subunit in addition to the 5 subunits identified in the screen.

## Improved RNAi vectors reveal distinct contributions of SWI/SNF subunits to AC invasion

Though many SWI/SNF assemblies have been described in mammalian and other systems, including BAF, PBAF, esBAF, GBAF, nBAF, and npBAF [53], to date, BAF and PBAF are the only SWI/SNF assemblies that have been described in *C. elegans*. Both assemblies consist of core subunits (SWSN-1, SWSN-4, SWSN-5) and accessory subunits (DPFF-1, SWSN-2.1/HAM-3, SWSN-2.2, SWSN-3, SWSN-6, and PHF-10), collectively referred to as common factors [47, 54]. These common factors are bound by assembly-specific subunits in a mutually exclusive manner, which confers the distinct character of each of the two assemblies (**Fig 2A**). Due to the absence of thorough biochemical investigation into the SWI/SNF complex in *C. elegans*, previous publications have classified subunits as part of the SWI/SNF core, accessory, or BAF/PBAF assemblies based on homology and phenotypic analyses [30, 36, 44, 55]. The prevailing model for the two SWI/SNF assemblies in *C. elegans* is that either the SWSN-8 subunit associates with common factors to form the BAF assembly, or the SWSN-7, SWSN-9, and PBRM-1 subunits associate with common factors to form the PBAF assembly [44, 55, 56]. Prior investigations into SWI/SNF have revealed a wide array of developmental contexts in which the BAF and PBAF assemblies have overlapping and distinct roles in the regulation of cell cycle control, differentiation, and differentiated behavior [30, 36, 55, 57–61].

To investigate the contribution of individual SWI/SNF subunits to AC invasion and to distinguish potentially distinct roles of the BAF and PBAF assemblies, we generated improved RNAi constructs utilizing the T444T vector [62] to target representative subunits of the core and both SWI/SNF assemblies (**S3 Table**). Knockdown of SWI/SNF subunits in whole-body RNAi sensitive animals following treatment with T444T RNAi vectors resulted in penetrant loss of invasion. The majority of ACs failed to invade following treatment with RNAi targeting the core SWI/SNF ATPase subunit *swsn-4* or core subunit *swsn-1* (90% and 94%, respectively; n = 50 animals) (**Fig 2B and 2E**). Knockdown of subunits specific to either SWI/SNF assembly resulted in a lower penetrance of AC invasion defects. RNAi-mediated knockdown of the BAF assembly subunit *swsn-8* resulted in loss of AC invasion in 32% of treated animals (n = 50 animals) (**Fig 2C and 2E**). Knockdown of the PBAF assembly subunits with *pbrm-1(RNAi)*, *swsn-7(RNAi)*, or *swsn-9(RNAi)* resulted in a less penetrant loss of AC invasion (18%, 20%, and 12%, respectively; n = 50 animals) (**Fig 2D and 2E**). Importantly, across all RNAi treatments targeting individual SWI/SNF subunits, at least one cell in the ventral uterus dorsal to the primary vulva expressed the fluorescent AC reporter, suggesting that loss of the SWI/SNF complex does not compromise AC specification.

Interestingly, in addition to a single non-invasive AC phenotype, RNAi-mediated knockdown of *swsn-1*, *swsn-4* or *swsn-8* also resulted in a second phenotype characterized by multiple uterine cells expressing the AC reporter (*cdh-3p*::*mCherry*::*moeABD*) which failed to invade the BM (*laminin*::*GFP*) (32%, 30% and 8%, respectively) (**Fig 2B, 2C and 2E**). In all instances where more than one cell expressed the AC reporter, no breach in the underlying

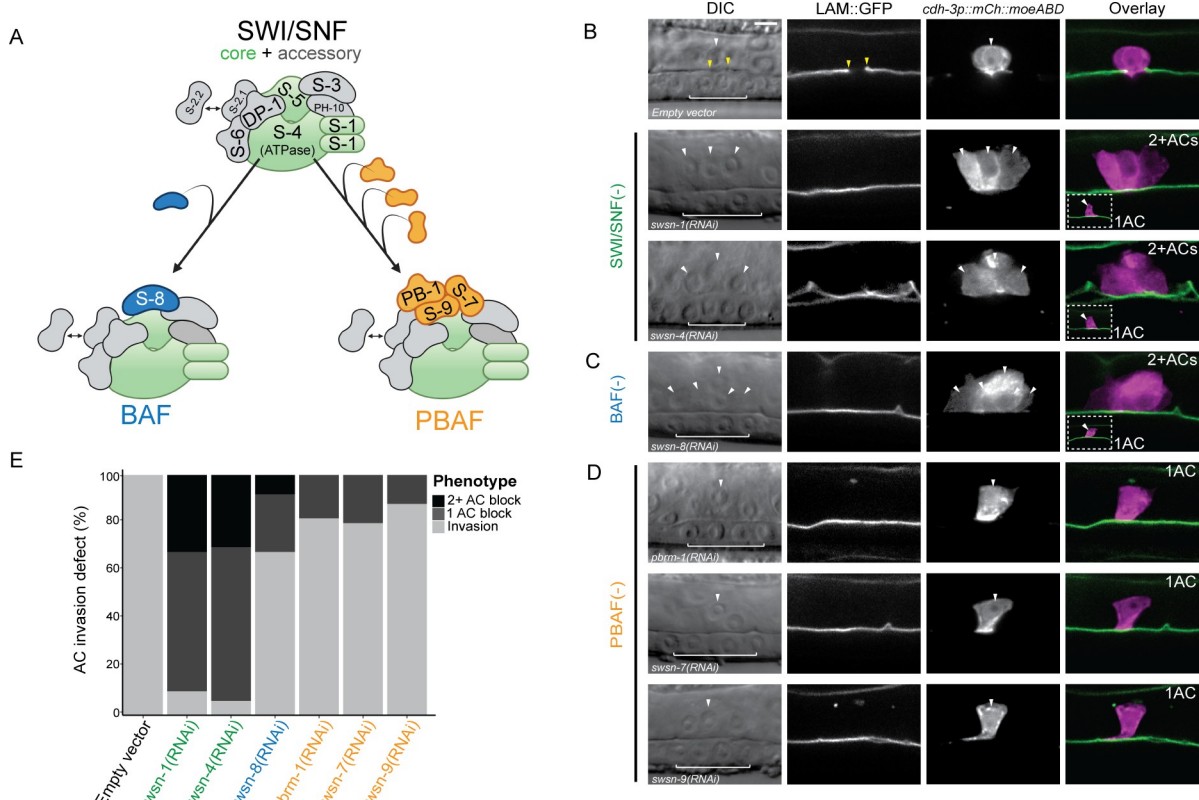

**Fig 2. Enhanced RNAi targeting SWI/SNF core, BAF, and PBAF subunits results in penetrant invasion defects. (A)** Schematic depicting the *C. elegans* SWI/SNF common factors (core and accessory subunits, top), along with BAF (left, blue), and PBAF (right, orange) assemblies. **(B-D)** DIC (left), corresponding fluorescence images (middle), and fluorescence overlay (right) representing loss of AC (magenta, *cdh-3p::mCherry::moeABD*) invasion through the BM (green, *laminin::GFP*) following RNAi depletion of SWI/SNF core (*swsn-1* and *swsn-4*) **(B)**, BAF (*swsn-8*) **(C)**, and PBAF (*pbrm-1*, *swsn-7*, and *swsn-9*) **(D)** subunits. White arrowheads indicate ACs, yellow arrowheads indicate boundaries of breach in the BM, and white brackets indicate 1° VPCs. In cases where multiple cells expressed the AC reporter (2+ACs) in the same animal following RNAi treatment, each cell expressing the AC reporter is indicated with a white arrowhead. In cases where multiple cells expressed the AC reporter (2+ACs), a representative image from the same treatment of a single AC that fails to breach the BM is displayed as an inset (white dashed box, bottom left). Scale bar, 5 μm. **(E)** Stacked bar chart showing the penetrance of AC invasion defects following treatment with SWI/SNF RNAi depletion, binned by AC phenotype (n≥50 animals examined for each treatment).

BM was detected at the P6.p 4-cell stage. In contrast, only the single non-invasive AC phenotype resulted from RNAi treatment targeting PBAF assembly subunits (**Fig 2D and 2E**). These results suggest that the SWI/SNF assemblies BAF and PBAF may promote AC invasion through distinct mechanisms, perhaps via regulation of both a cell cycle-dependent and -independent mechanism, respectively.

## Characterization of endogenous GFP reporter alleles and the efficacy of improved SWI/SNF RNAi vectors

Next, to confirm expression of SWI/SNF subunits in the AC and to quantitatively assess the potency of our enhanced SWI/SNF RNAi vectors, we utilized CRISPR/Cas9 genome engineering to generate GFP-tagged alleles of *swsn-4* and *swsn-8*, inserting a codon-optimized GFP tag into the 5' end and 3' end of the *swsn-4* and *swsn-8* loci, respectively (**Fig 3A and 3B,** top) [63]. The GFP-tagged endogenous strains showed ubiquitous and nuclear-localized expression of GFP::SWSN-4 and SWSN-8::GFP throughout the *C. elegans* developmental life cycle (**Fig 3A and 3B**, bottom). We also obtained a strain containing an endogenously eGFP-labeled PBAF

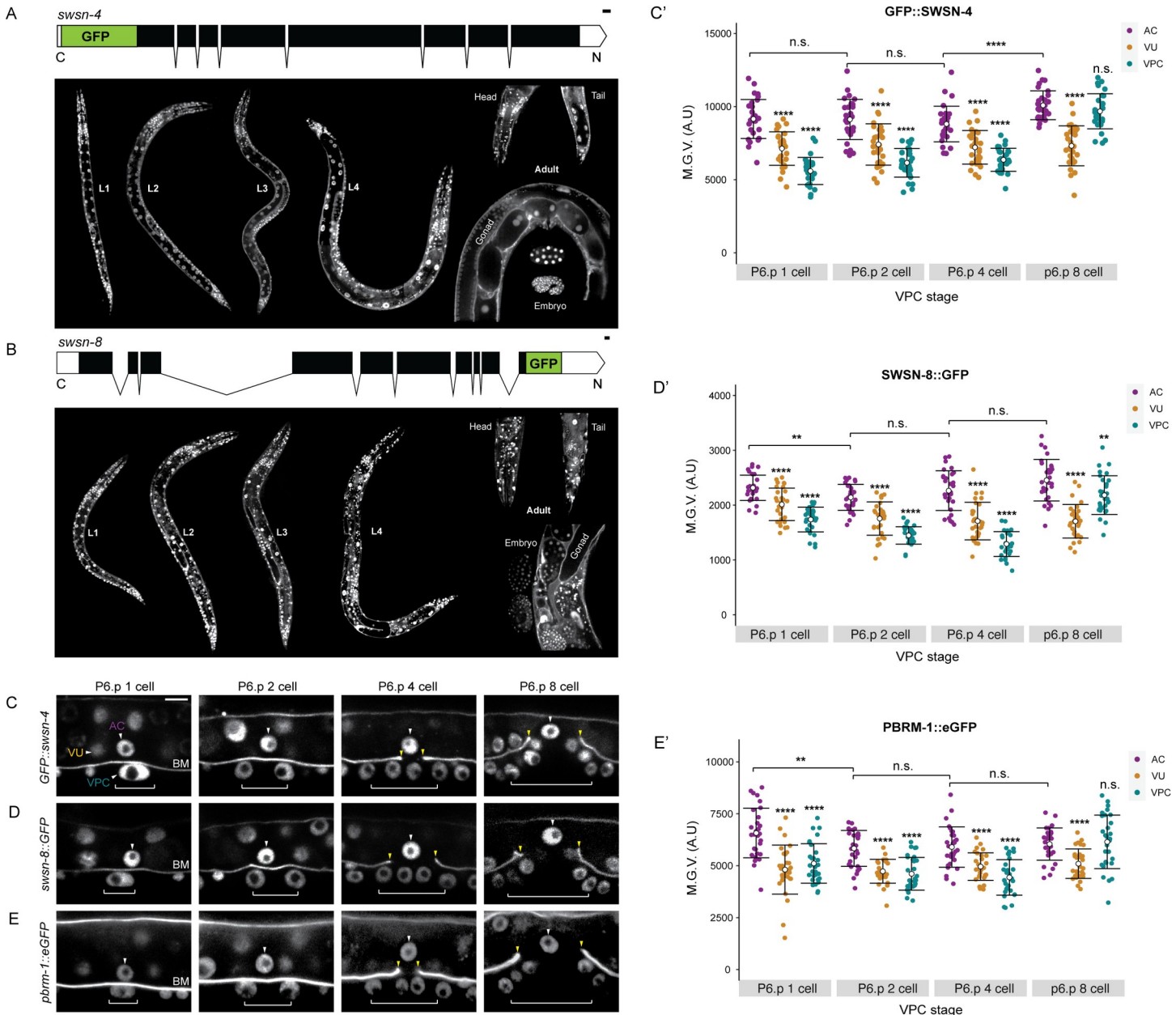

**Fig 3. SWI/SNF fluorescent knock-ins express in the AC pre-, during, and post-invasion.** Schematics (from http://wormweb.org/exonintron) depicting GFP insertion into the endogenous N and C termini of *swsn-4* (**A**, top) and *swsn-8* (**B**, top), respectively. Scale bar, 100bp. (**A-B**, bottom) Fluorescent micrographs depicting protein expression of each SWI/SNF subunit and BM (*laminin::GFP*) in all larval stages (L1-L4), adult, and embryos. Images scaled for clarity. Fluorescent micrographs depicting expression of GFP::SWSN-4 (**C**), SWSN-8::GFP (**D**), and PBRM-1::eGFP (**D**) in the AC, VU, and VPCs from the P6.p 1-cell to the P6.p 8-cell stages of development. White arrowheads indicate AC, white brackets indicate 1° VPC stage. Scale bar, 5μm (**C'-E'**) Quantification of endogenous GFP expression of SWI/SNF subunit in the AC, VU, and VPC over time. Statistical comparisons were made for the expression of each SWI/SNF subunit in the AC over time (asterisks or n.s. above black brackets) or between the expression of each subunit in the AC relative to the expression of the same subunit in the neighboring VPCs or VUs at the same time (asterisks or n.s. below black brackets) using Student's *t*-test (n≥30 for each stage and subunit; p values are displayed above compared groups). n.s. not significant.

subunit (*pbrm-1::eGFP*) from the *Caenorhabditis* Genetics Center (CGC). We quantified fluorescence protein expression of SWI/SNF core ATPase (GFP::SWSN-4), BAF (SWSN-8::GFP), and PBAF (PBRM-1::eGFP) subunits in the AC during vulval development across the L3 and early L4 stages, as defined by the division pattern of the 1°-fated VPCs [14] (n≥28 animals per

stage) (**Fig 3C, 3D, 3E, 3C', 3D' and 3E'**). Expression of all three subunits was enhanced in the AC relative to the neighboring ventral uterine (VU; *swsn-4*: 18%, *swsn-8*: 21%, *pbrm-1*: 17% enhanced) and 1˚ VPC (*swsn-4*: 30%, *swsn-8*: 38%, *pbrm-1*: 23% enhanced) lineages during AC invasion (P6.p 2-cell– 4-cell stage) (**Fig 3C', 3D' and 3E'**). Late in vulval development at the P6.p 8-cell stage, expression of GFP::SWSN-4 and PBRM-1::eGFP increases in the 1˚ VPCs and is no longer statistically separable from expression in the AC (**Fig 3C' and 3E'**), whereas expression of SWSN-8::GFP in the VPCs also increases but remains significantly lower than in the AC (**Fig 3B'**).

We treated SWI/SNF endogenously labeled GFP-tagged strains with our improved RNAi vectors to precisely quantify the efficiency of RNAi-mediated knockdown of target SWI/SNF complex subunits and to correlate this loss with the resulting AC phenotypes. Treatment with either *swsn-4(RNAi)* or *swsn-8(RNAi)* vectors resulted in robust depletion of fluorescence expression of GFP::SWSN-4 (94% depletion) and SWSN-8::GFP (81% depletion) in the AC (**S2A, S2B, and S2D Fig**) and penetrant loss of invasion (90% and 30%, respectively; n = 30 animals for each condition) (**S2E Fig**). We also noted instances where multiple cells expressed the AC reporter (23% and 10%, respectively; n = 30 animals for each condition) (**S2E Fig**). Treatment of the PBRM-1::eGFP strain with *pbrm-1(RNAi)* revealed weaker but significant knockdown of PBRM-1 protein (49% depletion) (**S2D Fig**), and a lower penetrance of invasion defects (17%; n = 30 animals) (**S2C, S2D and S2E Fig**). It is unclear why PBRM-1::eGFP endogenous protein level in the AC of animals treated with enhanced *pbrm-1(RNAi)* remains considerably higher compared to treatment of strains containing *swsn-4* or *swsn-8* reporter alleles with their respective RNAis. We hypothesize that this may be the consequence of differential protein perdurance of the PBRM-1 protein. We note that the strength of the RNAi clones as determined by quantitative fluorescence analysis of RNAi-treated endogenous SWI/SNF::GFP strains tracks with the relative penetrance of AC invasion defects we observed in both the whole body sensitive RNAi strain (**Fig 2E**) and the endogenous strains themselves (**S2E Fig**). Altogether, these results confirm the dynamic expression of the SW/SNF core, BAF, and PBAF subunits in the AC before, during, and after invasion and demonstrate the effectiveness of our improved SWI/SNF-targeting RNAi vectors.

## *C. elegans* SWI/SNF subunits exhibit intra-complex and low levels of inter-assembly cross-regulation

Work in cell culture has revealed that the mammalian SWI/SNF (mSWI/SNF) complex is assembled in a step-wise fashion, with stability of the complex as a whole and association of individual subunits depending on the prior expression and association of other subunits [64]. To date it is unknown whether in *C. elegans* individual SWI/SNF subunits activate other SWI/SNF subunits. It is also unclear whether subunits of the two assemblies in *C. elegans*–BAF and PBAF–stabilize the core protein subunits or vice-versa. Therefore, we used our endogenously labeled SWI/SNF::GFP strains to ask whether transcriptional knockdown of individual subunits of the core, BAF, or PBAF induce changes in protein expression of other subunits at the time of AC invasion (**S3 Fig**).

First, to determine whether representative subunits of the SWI/SNF assemblies promote or stabilize the ATPase of the complex, we treated *GFP*::*swsn-4* animals with either *swsn-8(RNAi)* or *pbrm-1(RNAi)* (**S3A Fig**). Quantification of fluorescence expression in AC nuclei of *swsn-8 (RNAi)* treated animals at the P6.p 4-cell stage revealed significantly lower GFP::SWSN-4 levels relative to the control group (34% GFP::SWSN-4 depletion) (**S3A and S3D Fig**). RNAi knockdown of the PBAF subunit *pbrm-1* also resulted in a significant but weaker loss of ATPase expression in the AC (11% GFP::SWSN-4 depletion) (**S3A and S3D Fig**). These results suggest

that individual subunits of either SWI/SNF assembly exhibit inter-complex regulation and may contribute to the protein stability and/or expression of the SWI/SNF ATPase in the *C. elegans* AC, with the BAF complex playing a potentially dominant activating role with respect to the ATPase.

Next, we treated animals containing either the *swsn-8* or *pbrm-1* endogenous GFP-reporters with enhanced RNAi to knockdown the expression of the SWI/SNF ATPase or the representative subunit of the alternative SWI/SNF assembly. Interestingly, while unaffected by knockdown of the PBAF assembly subunit *pbrm-1*, RNAi knockdown of the ATPase *swsn-4* resulted in a 42% increase in the expression of SWSN-8::GFP in the AC (**S3B**, **S3C and S3D Fig**). Finally, relative to the expression of the endogenous PBAF subunit in the ACs of control animals, AC nuclei of PBRM-1::eGFP animals treated with *swsn-4(RNAi)* had significantly lower levels of protein expression (38% PBRM-1::eGFP depletion), whereas ACs in *swsn-8 (RNAi)* treated animals expressed 13% more PBRM-1::eGFP (**S3C and S3D Fig**).

Since knockdown of either *swsn-4* or *swsn-8* subunits resulted in two distinct AC phenotypes–individual animals with single non-invasive ACs and animals with multiple non-invasive cells expressing the AC-reporter—we next sought to determine whether these two phenotypes were distinct with respect to SWI/SNF subunit expression. To do this, we binned data from the intra-complex RNAi experimental series (**S3A, S3B**, **S3C and S3D Fig**) into the two non-invasive phenotypes and compared the fluorescence expression levels of the endogenous proteins within SWI/SNF RNAi conditions. Given the infrequency of the multi-AC phenotype, statistical comparisons were necessarily limited to treatments in which the population of animals contained at least 10 multi non-invasive AC events. Treatment of SWSN-8::GFP with *swsn-4(RNAi)* resulted in a total of 24 multi non-invasive ACs (53 ACs total; n = 41 animals) and no significant difference was detected in SWSN-8::GFP expression between the nuclei of the single non-invasive AC phenotype and the multi non-invasive AC phenotype groups (**S3E Fig**). The second statistical comparison was made between the two phenotypes in PBRM-1::eGFP animals treated with *swsn-8(RNAi)* (**S3E Fig**), in which 14 multi non-invasive ACs were detected (51 ACs total; n = 42 animals). Quantification of endogenous PBRM-1:: eGFP fluorescence expression in this condition revealed a slight (12%) increase in expression of the PBAF subunit in the nuclei of ACs of the multi non-invasive phenotype group compared to the single non-invasive phenotype (**S3E Fig**), reflecting the general increase in PBRM-1 levels detected in the non-binned data (**S3D Fig**).

Based on these results, a tentative model for epistatic interactions between the SWI/SNF ATPase, BAF, and PBAF assembly subunits can be composed for the AC (**S3F Fig**). Our data indicate that some degree of SWI/SNF intra-complex and inter-assembly regulation occurs in the AC. We find that the most significant aspect of SWI/SNF intra-complex regulation is exercised by the ATPase on the assembly specific subunits, where *swsn-4* knockdown results in a significant increase in BAF/SWSN-8 and a significant decrease PBAF/PBRM-1. SWI/SNF inter-assembly regulation appears to be weaker in the AC as knockdown of *pbrm-1* does not affect SWSN-8::GFP expression, and knockdown of *swsn-8* results in a slight increase in PBRM-1::GFP expression.

## The SWI/SNF ATPase SWSN-4 provides dose-dependent regulation of AC invasion

The degree to which the SWI/SNF complex contributes to tumorigenesis in clinical settings has been linked to the dose of functional SWI/SNF ATPase in precancerous and transformed cells [33, 35, 65]. Previous work in *C. elegans* has demonstrated a similar dose dependent relationship between SWI/SNF and cell cycle control [36]. Additionally, our results with enhanced

SWI/SNF RNAi across SWI/SNF::GFP endogenous strains (**S2 Fig**) suggest that stronger knockdown of SWI/SNF subunits may correlate with an increased penetrance of invasion defects. To determine whether the phenotypic dosage sensitivity seen in cancer and *C. elegans* mesodermal (M) cell development is indeed characteristic of SWI/SNF in the promotion of AC invasion [31], we modulated expression of GFP::SWSN-4 using a combination of RNAi-mediated knockdown and AC-specific GFP-targeting nanobody technology.

Though RNAi treatment targeting the *swsn-4* subunit in the endogenously-tagged strain resulted in significant knockdown of fluorescence expression of GFP::SWSN-4 in the AC, some loss of expression was noted in other tissues in treated animals, including the 1˚ VPCs, which contribute to AC invasion non-autonomously [14, 66] (**S2 Fig**). Thus, to limit loss of expression to the AC, we used an anti-GFP nanobody fused to a SOCS-box containing a ubiquitin ligase adaptor, driven with tissue-specific promoters to achieve lineage-restricted protein depletion [31] (**Fig 4**). To follow the expression of the anti-GFP nanobody transgenes, we also included a fluorescent histone label separated from the anti-GFP nanobody sequence by the p2a viral self-cleaving peptide (*ACp*::*antiGFP-nanobody*::*p2a*::*his-58*::*mCherry*). We generated two anti-GFP nanobody constructs, using conserved *cis*-regulatory elements from the

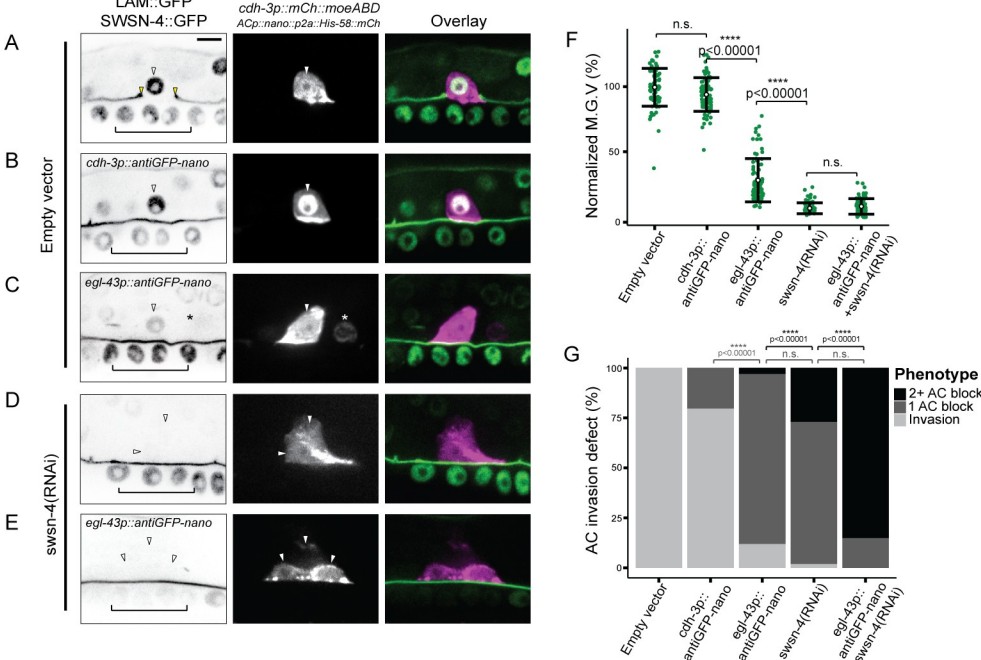

**Fig 4. AC invasion and cell cycle arrest depend on dosage of SWI/SNF ATPase. (A-E)** Representative fluorescence images depicting expression of BM marker (*laminin*::*GFP*) and endogenous GFP::SWSN-4 (left), AC reporter (*cdh-3p*::*mCherry*::*moeABD*, middle), and fluorescence overlay (right) across experimental treatments. White arrowheads indicate ACs, yellow arrowheads in A indicate boundaries of breach in BM. Black brackets indicate 1˚ VPCs. In cases where multiple cells expressed the AC reporter in the same animal, each is indicated with a single white arrowhead. Asterisk indicates anti-GFP nanobody expression in neighboring VU cell. **(F)** Quantification of mean gray values (M.G.V.) of endogenous GFP::SWSN-4 in ACs in control animals (empty vector) and across all experimental treatments normalized to mean fluorescent expression in wildtype animals (n≥40 animals per treatment, p values for Student's *t*-test comparing expression of successive knockdown are displayed on the figure). In this and all other figures, open circles and error bars denote mean±standard deviation (s.d.). n.s. not significant. **(G)** Stacked bar chart showing quantification of AC invasion defects corresponding to each treatment, binned by AC phenotype (n≥40 animals per condition; p values for Fisher's exact test comparing phenotypes of successive knockdown strategies are displayed above compared groups). Grey brackets indicate statistical significance between invasion total in each condition compared to invasion defect total. Black brackets indicate statistical significance between incidences of invasion defects with multiple ACs compared to incidences of invasion defects with single ACs. n.s. not significant.

*cdh-3* and *egl-43* promoters [22, 24, 40, 67, 68] and introduced them into a strain containing the endogenous *GFP::swsn-4* allele as well as AC and BM reporters (**Fig 4B and 4C**). The *cdh-3*-driven nanobody transgene (*cdh-3p::antiGFP-nanobody::p2a::his-58::mCherry*) resulted in a weak reduction of GFP::SWSN-4 levels with no significant difference in fluorescence expression in the AC compared to wildtype animals (6% depletion; n = 80 animals) (**Fig 4F**); however, consistent with the wildtype expression of the *cdh-3* promoter [22, 40], it expressed specifically in the AC and resulted in defective AC invasion, suggesting partial loss of function (21% AC invasion defect; n = 102) (**Fig 4B and 4G**). The *egl-43p::antiGFP-nanobody* transgene (*egl-43p::antiGFP-nanobody::p2a::his-58::mCherry*) expression pattern was also consistent with the wildtype expression characterized in previous work [22, 67–69], as indicated by nuclear expression of HIS-58::mCherry in the AC and in the neighboring ventral uterine and dorsal uterine (VU/DU) cells (**Fig 4C**; asterisk denotes HIS-58::mCherry expression in a non-AC ventral uterine cell) [22, 40, 67]. Importantly, as the AC invades independent of VU/DU cells [14], anti-GFP expression in these tissues should not affect AC invasion. Similar to animals treated with *swsn-4(RNAi)* (**Fig 4D**), *egl-43p::antiGFP-nanobody*-mediated protein depletion of GFP::SWSN-4 resulted in a significant loss of fluorescence expression in the AC (71% GFP depletion; n = 80 animals) (**Fig 4C and 4F**) as well as a penetrant loss of invasion and incidence of individual animals with multiple uterine cells that were in contact with the ventral BM and expressed the AC reporter (88% AC invasion defect, 3% multiple AC phenotype; n = 101 animals) (**Fig 4G**). These results support our uterine-specific SWI/SNF RNAi results and provide strong evidence for a cell-autonomous role for the SWI/SNF complex in promoting cell invasion and cell cycle arrest.

To further deplete *swsn-4* expression in the AC, we treated transgenic *egl-43p::antiGFP-nanobody* animals with *swsn-4(RNAi)* (**Fig 4E**). Strikingly, in this combination knockdown strategy, the AC invasion defect was completely penetrant and the frequency of multiple cells expressing the AC specification reporter drastically increased relative to treatment with *swsn-4 (RNAi)* or the *egl-43*-driven anti-GFP nanobody conditions alone (83% multiple AC phenotype; n = 41 animals) (**Fig 4G**). Together, these results demonstrate a phenotypic spectrum that corresponds to successive loss of *swsn-4* in the AC. Moderate loss of the ATPase results in single non-invasive ACs in animals containing *cdh3p::antiGFP-nanobody*. Strong loss of expression in the *egl-43p::antiGFP-nanobody* background or following treatment with *swsn-4 (RNAi)* results in animals with both single and multiple non-invasive ACs. Finally, in the strongest knockdown condition–*egl-43p::antiGFP-nanobody* animals treated with *swsn-4 (RNAi)*—multiple non-invasive ACs were present per animal with near complete penetrance. Though the combination of *swsn-4(RNAi)* and antiGFP-nanobody-mediated depletion resulted in robust loss of expression of the core ATPase of the SWI/SNF complex, the fluorescence expression was not significantly different than treatment with *swsn-4(RNAi)* alone (93% vs. 92% GFP depletion, respectively; n≥41 animals for each treatment) (**Fig 4F**). We theorize that in these conditions, the fluorescence values were beyond our threshold ability to quantify based on the fluorescence detection limits of our imaging system. Altogether, these data demonstrate that in the AC, the ATPase of the SWI/SNF complex contributes to invasion cell-autonomously and in a dose-dependent manner.

## Improved *swsn-4(RNAi)* vector is sufficient to recapitulate a null phenotype in the M lineage

A recent study focusing on cell cycle control of SWI/SNF throughout *C. elegans* muscle and epithelial differentiation demonstrated tissue and lineage-specific phenotypes following weak or strong loss of core SWI/SNF subunits [36]. Within the M lineage that gives rise to posterior

body wall muscles (BWMs), coelomocytes (CCs), and reproductive muscles or sex myoblast (SMs) descendants, different cell types responded differently to loss of SWI/SNF. In the BWM, strong loss of SWI/SNF resulted in hyperproliferation, like the phenotype we detect in the AC. The opposite is true in the SM lineage, where modest knockdown of *swsn-4* resulted in hyperproliferation while complete loss of *swsn-4* expression resulted in a null phenotype where SMs failed to divide and arrest in S phase [36]. We next sought to validate the strength of our enhanced *swsn-4(RNAi)* vector by examining the SM proliferative state. To accomplish this, we treated animals containing a lineage-restricted cyclin-dependent kinase (CDK) activity sensor (*unc-62p*::DHB::2xmKate2) with *swsn-4(RNAi)* (**S4A Fig**). In this genetic background, we determined the number (**S4B Fig**) and cell cycle state (**S4C Fig**) of SM cells at a time when the majority of SMs in control animals had finished cycling and subsequently differentiated (late P6.p 8-cell stage; 16 SM cell stage). Animals treated with *swsn-4(RNAi)* had significantly fewer SM cells than controls (mean SMs/animals = 5; n = 31 animals) (**S4B Fig**) with many instances of SMs that failed to enter a single round of cell division (n = 20 single SMs out of 43 animals). Interestingly, 28% (12/43) of animals treated with *swsn-4(RNAi)* were absent of SMs on either the left or the right side, whereas 100% (30/30) control animals had SMs on both sides, which may indicate a defect in specification, early cell division, and/or migration of SMs. To quantify cell cycle state, we measured localization of an SM-specific CDK sensor, which uses a fragment of mammalian DNA Helicase B (DHB) fused to two copies of mKate2 [37, 70]. In cells with low CDK activity that are quiescent or post-mitotic, the ratiometric CDK sensor is strongly nuclear localized [37, 68, 70]. In cycling cells with increasing CDK activity, the CDK sensor progressively translocates from the nucleus to the cytosoplasm, with a ratio approaching 1.0 in S phase and >1 in cells in $G_2$ [37]. Thus, the cytoplasmic:nuclear (C/N) ratio of DHB::2xmKate2 can serve as a proxy to identify cell cycle state. By the time the majority of SMs in the control condition were differentiating and arrested in a $G_0$ cell cycle state (mean C/N ratio = 0.320; n = 90 SMs) (**S4C Fig**), many animals treated with *swsn-4(RNAi)* had single SMs that failed to divide and a mean DHB C/N ratio indicative of a long pause or arrest in S phase [37] (Avg. C/N ratio = 0.803; n = 20 SMs) (**S4C Fig**). Together, these results suggested that the strength of our enhanced *swsn-4(RNAi)* targeting vector is sufficient to recapitulate a *swsn-4* null condition in the SM lineage, as we detected both the hypoproliferative phenotype and S-phase arrest that was observed using a lineage-restricted catalytically inactive SWI/SNF ATPase [36].

## The BAF assembly contributes to AC invasion via regulation of $G_0$ cell cycle arrest

Having established that strong depletion of the SWI/SNF complex results in a fully penetrant defect in AC invasion with a high percentage of individual animals possessing multiple non-invasive ACs (**Fig 4**), we next investigated whether the extra ACs observed were the consequence of inappropriate AC proliferation [22, 68]. To determine whether the SWI/SNF complex is required for $G_0/G_1$ cell cycle arrest in the AC, we quantified CDK activity in the AC using a ubiquitously expressed *rps-27p*::DHB::GFP transgene paired with AC (*cdh-3p*::*mCherry*::*moeABD*) and BM (*laminin*::*GFP*) reporters in live animals following RNAi-mediated knockdown of SWI/SNF core (*swsn-4*), BAF (*swsn-8*), and PBAF (*pbrm-1*) subunits (**Fig 4**). In wild-type invasive ACs, we observed strong nuclear localization of the CDK sensor and quantified a cytoplasmic/nuclear (C/N) ratio indicative of $G_0/G_1$ arrest (mean C/N ratio: 0.226 +/-0.075, n = 67 animals) (**Fig 5A and 5F**). To distinguish whether the wild-type AC C/N ratio is actually indicative of $G_0$ rather than $G_1$ arrest, we quantified the CDK activity in the neighboring uterine Pi cells at the P6.p 8-cell 1° VPC stage following their terminal division to

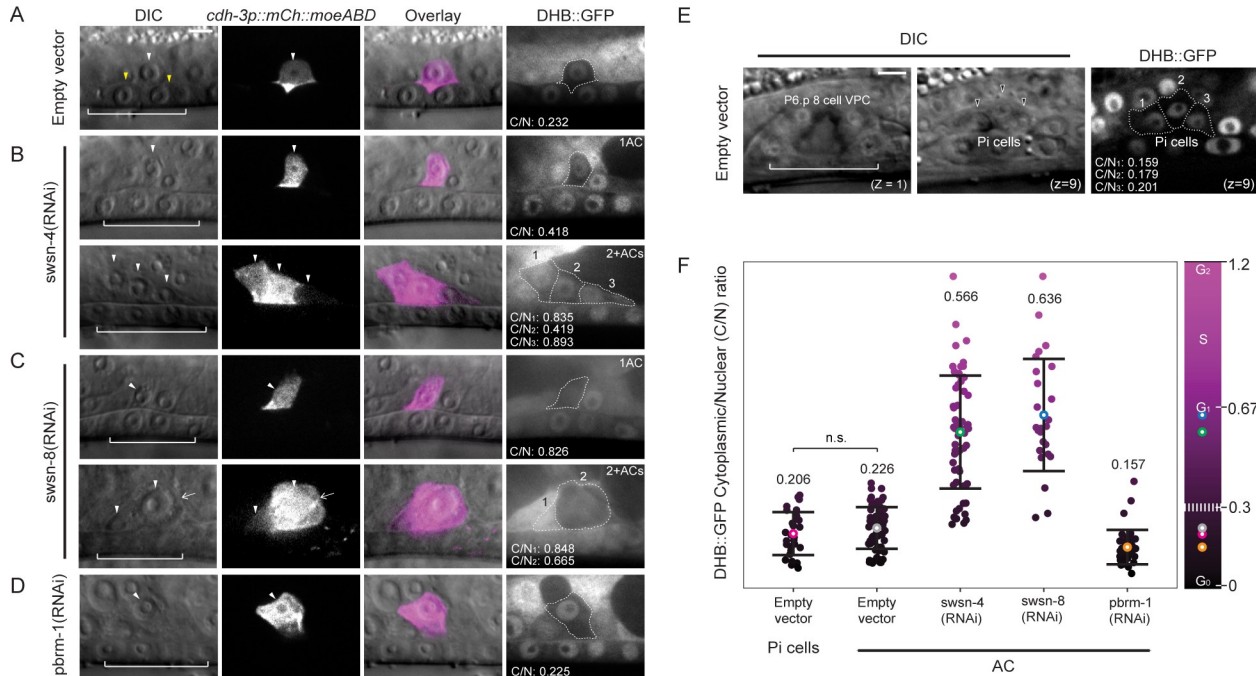

**Fig 5. CDK sensor reveals SWI/SNF contribution to $G_0$ arrest in the AC.** Micrographs depicting DIC (left), AC (*cdh-3p::mCherry::moeABD*, center-left), DIC overlay (center-right), and DHB-based CDK activity sensor (right) in empty vector **(A)** and following treatment with SWI/SNF RNAi targeting subunits of the core (*swsn-4*, **B**), BAF (*swsn-8*, **C**) and PBAF (*pbrm-1*, **D**) assemblies. White arrowheads indicate ACs, yellow arrowheads in A indicate boundaries of breach in BM, and white brackets indicate 1° VPCs. In cases where treatment resulted in multiple cells expressing the AC reporter in the same animal, representative images of both single (1AC, top) and mitotic (2+ACs, bottom) phenotypes are given, and each AC is indicated with a single white arrowhead. Quantification of the cytoplasmic:nuclear (C/N) ratio of DHB::GFP in ACs (white dotted outline) is listed in the bottom left of each panel. Mitotic ACs are numbered, and C/N ratios are provided for each **(B-C)**. White arrow in C indicates an AC that is out of the focal plane. **(E)** Representative single z-plane micrographs of the vulva at the P6.p 8-cell stage (left, z = 1) and the terminal Pi cells (middle, z = 9) in DIC, and DHB-based CDK activity sensor in Pi cells (right). Quantification of the C/N ratio of DHB::GFP in three of four Pi cells (white dotted outline) that are in the plane of the image is listed in the bottom left. **(F)** Quantification of C/N DHB::GFP ratios for wild-type terminally divided Pi cells and all ACs in empty vector control and each RNAi treatment (n≥30 animals per treatment). Statistical comparison was made between the mean C/N ratio of ACs in control (empty vector) compared to control (empty vector) Pi cells using Student's *t*-test (n≥30 for each stage and subunit; p values are displayed above compared groups). Mean C/N ratio is represented by colored open circles and correspond to numbers above the data. Gradient scale depicts cell cycle state as determined by quantification of each Pi cell or AC in all treatments (n≥30 animals per treatment), with dark/black depicting $G_0$ and lighter/magenta depicting $G_2$ cell cycle states. Dashed white line on gradient scale bar (right) corresponds to boundaries between $G_0$ and $G_1$ phases. Colored open circles on the gradient scale correspond to the mean C/N ratio in each of the same color. n.s. not significant.

establish a $G_0$ reference point [71, 72] (mean C/N ratio: 0.206+/-0.078, n = 30 animals) (**Fig 5E and 5F**). We found no significant difference between the CDK activity of terminal Pi cells and wild-type invading ACs, suggesting that the wild-type AC exists in a $CDK^{low}$ $G_0$, pro-invasive state (**Fig 5A, 5E and 5F**). In animals treated with *pbrm-1(RNAi)*, the CDK sensor also localized principally in the nucleus of ACs that failed to invade (mean C/N ratio: 0.157+/-0.063, n = 41 animals) and only a single non-invasive AC was observed per animal (**Fig 5D and 5F**). In contrast, following treatment with *swsn-8(RNAi)*, the majority of ACs that failed to invade the BM were in the $G_1$/S phases of the cell cycle (mean C/N ratio: 0.636+/-0.204, n = 21 animals) (**Fig 5C and 5F**). Finally, like the *swsn-8(RNAi)* condition, loss of expression of the core ATPase of the SWI/SNF complex through treatment with *swsn-4(RNAi)* resulted in a broad range of C/N ratios (C/N ratio min: 0.240, C/N ratio max: 1.140, mean C/N ratio: 0.566 +/-0.205; n = 40 animals) in animals with single or multiple non-invasive ACs (**Fig 5B and 5F**). Interestingly, the *swsn-4(RNAi)* treatment resulted in a higher proportion of non-invasive $G_0$ phase (C/N ratio < 0.3) ACs (14%, n = 48 cells) than were present in the *swsn-8(RNAi)*

treated population (8%, n = 25) (**Fig 5B and 5C,** upper panel**).** These findings reemphasize the functional dependence of both SWI/SNF BAF and PBAF assemblies on the core ATPase of the complex, as the distribution of cell cycle states in ACs following *swsn-4(RNAi)* treatment represents both the cell cycle-dependent and cell cycle-independent phenotypes seen in ACs deficient in the *swsn-8* or *pbrm-1* subunits, respectively.

## Forced $G_0$ arrest through ectopic CKI-1 rescues invasive potential in BAF-deficient but not PBAF-deficient ACs

We have previously proposed and characterized a dichotomy that exists between invasion and proliferation in the AC [22, 24]. As evidence of this, loss of two of the three TFs that function in a cell cycle-dependent manner to maintain the AC in a cell cycle-arrested state (*nhr-67*/Tlx and *hlh-2*/Daughterless) can be rescued through induced expression of a cyclin dependent kinase inhibitor, *cki-1* (p21/p27) [22]. These results suggest that, at least in some cases, TF activity can be bypassed completely to promote AC invasion by maintaining $G_0$ arrest through direct cell cycle manipulation. To determine the extent to which the BAF assembly contributes to AC invasion through regulation of cell cycle arrest, we used a heat-shock inducible transgene to ectopically express CKI-1::mTagBFP2 in SWI/SNF-deficient ACs (**Fig 6**). Since the heat shock inducible transgene is ubiquitous and expresses variably between different animals and different tissues within an individual animal, we limited our analysis to animals with ACs with obvious mTagBFP2 fluorescence expression. While forced arrest in $G_0$ was insufficient to significantly rescue AC invasion in animals treated with *swsn-4(RNAi)* (**Fig 6B, 6B' and 6E**) or *pbrm-1(RNAi)* (**Fig 6D, 6D' and 6E**), ectopic *cki-1* (CKI-1::mTagBFP2) expression in the AC significantly rescued cellular invasion in animals treated with *swsn-8(RNAi)* (**Fig 6C, 6C' and 6E**). Strikingly, in 86% (6/7) of cases where ACs had proliferated prior to ectopic CKI-1 expression, forced $G_0$ arrest led to multiple ACs breaching the BM (**Fig 6F**), a phenotype we have reported previously using CKI-1 overexpression paired with loss of NHR-67. This demonstrated that mitotic ACs maintain the capacity to invade if they are re-arrested into a $G_0$ state, even in the absence of the SWI/SNF BAF subunit [24]. To corroborate our CKI-1 heat shock data, we used an AC-specific CKI-1 transgene (*cdh-3p::CKI-1::GFP*) to induce $G_0$ cell cycle arrest in *swsn-4-* and *swsn-8-* depleted ACs (**S5 Fig**). Similar to the heat shock results, lineage-restricted expression of CKI-1::GFP failed to rescue invasion in animals deficient in *swsn-4* (**S5A and S5B Fig**). However, transgenic *cdh-3p::CKI-1::GFP* animals treated with *swsn-8(RNAi),* had invasion defects significantly lower than control animals treated with *swsn-8(RNAi)* which lacked the $G_0$ rescue transgene (**S5B Fig**). Altogether, these data corroborate our DHB-based CDK sensor data (**Fig 4**), suggesting that the SWI/SNF assemblies differentially contribute to AC invasion with BAF specifically required for $G_0$ cell cycle arrest.

## SWI/SNF chromatin remodeling promotes the invasive GRN in the AC

Previous work has demonstrated that the gene regulatory network (GRN) that promotes AC invasion consists of both cell cycle-dependent and cell cycle-independent TF subcircuits [22, 68] (**Fig 1B**). In the cell cycle-dependent subcircuit of the TF-GRN, *egl-43* (EVI1/MEL), *hlh-2* (E/Daughterless), and *nhr-67* (TLX/Tailless) cooperate in a type 1 coherent feed-forward loop that is reinforced via positive feedback to retain the AC in a post-mitotic, invasive state [22, 68]. The cell cycle-independent subcircuit of the AC TF-GRN is governed by the *fos-1* (FOS) TF with feedback from both *egl-43* and *hlh-2* [22]. Since transcriptional knockdown of SWI/SNF ATPase results in both single and mitotic non-invasive AC phenotypes, we treated endogenously GFP-labeled strains for each TF in the GRN with the most penetrant enhanced SWI/SNF RNAi clone—*swsn-4(RNAi)*—to determine whether SWI/SNF chromatin remodeling

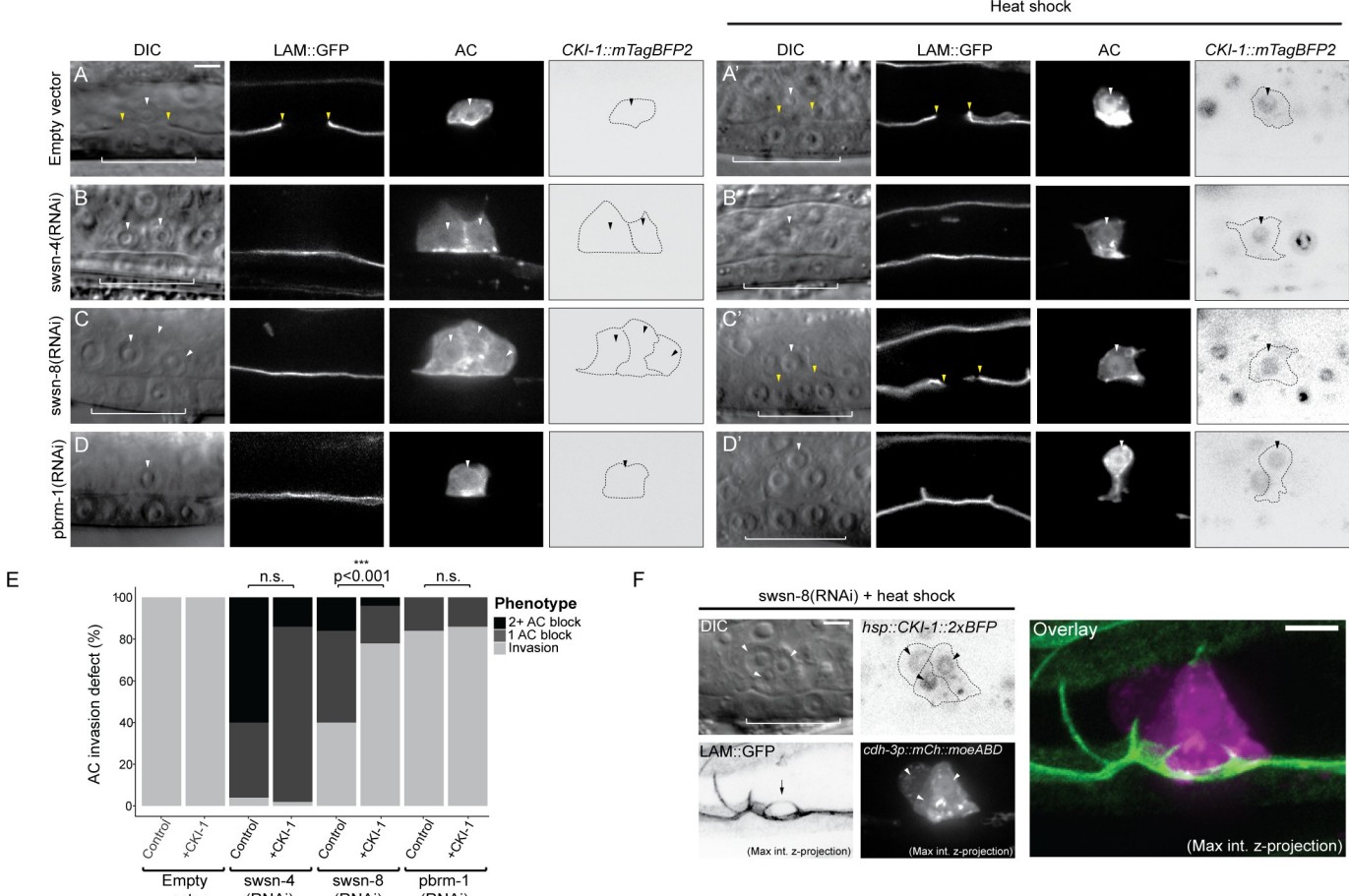

**Fig 6. BAF depletion is rescued by G₀ arrest.** Representative micrographs depicting DIC (left), BM (*laminin::GFP*, center-left), AC (*cdh-3p::mCherry::moeABD*, center-right), and CKI-1 (*hsp::CKI-1::mTagBFP2*) expression in empty vector control (**A-A'**) and treatment with SWI/SNF RNAi under standard conditions (**A-D**) and following heat shock induction of CKI-1 (**A'-D'**). CKI-1 images have been inverted for ease of visualization. White arrowheads indicate AC(s), yellow arrowheads indicate boundaries of breach in BM, and white brackets indicate 1° VPCs. Black dotted lines in *CKI-1*::mTagBFP2 panels delineate the boundaries of ACs; black arrowheads indicate position of nuclei in ACsScale bar, 5 μm. (**E**) Stacked bar chart showing percentage of AC invasion defects corresponding to each RNAi treatment under standard growth conditions (control) and following heat shock induction of CKI-1 (+CKI-1), binned by AC phenotype (n≥30 animals per condition; Fisher's exact test compared CKI-1(+) animals with control, non-heat shocked animals; p value is displayed above compared groups). n.s. not significant. (**F**) Representative micrographs of invasive group of *swsn-8* deficient ACs following induction of G₀/G₁ arrest. DIC (top-left), BM (bottom-left), CKI-1 expression (top-right), AC reporter (bottom-right). Max intensity z-projection of AC and BM reporter channels (right). Large breach in BM is indicated by black arrow in the bottom left panel. Scale bar, 5μm.

contributes to the regulation of either or both AC GRN subcircuits (**Fig 7**). In the cell cycle-dependent subcircuit, knockdown of the SWI/SNF ATPase resulted in significant loss of protein expression of EGL-43::GFP and NHR-67::GFP in the AC (39% and 26% GFP depletion, respectively; n≥41 animals) (**Fig 7A, 7C and 7E**). No significant difference was detected in the mean fluorescence expression of GFP::HLH-2 fusion protein in the AC upon knockdown of *swsn-4*, however the range of expression was broad following *swsn-4(RNAi)* treatment (~2% GFP increase; n≥50 animals) (**Fig 7B and 7E**). In the cell cycle- independent subcircuit, loss of the SWI/SNF complex following treatment of *fos-1*::*GFP* animals with *swsn-4(RNAi)* resulted in a more moderate depletion of expression in the AC (11% GFP depletion; n≥50 animals) (**Fig 7D and 7E**). While we cannot say whether SWI/SNF directly binds regulatory regions of these TFs, these results suggest that the SWI/SNF complex broadly remodels chromatin to promote both subcircuits of the pro-invasive AC GRN.

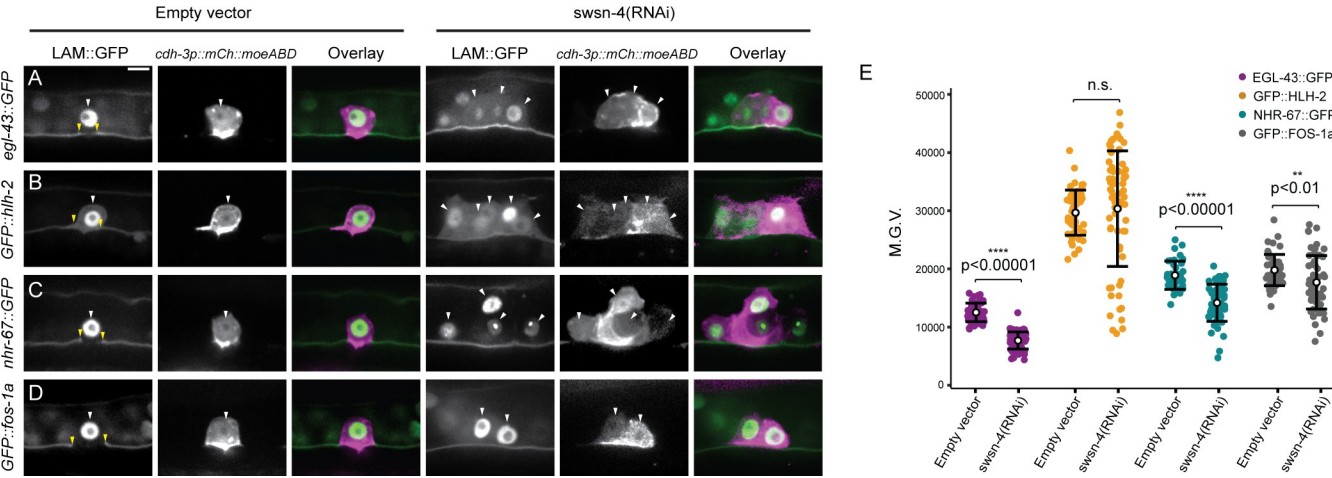

**Fig 7. SWI/SNF regulates TFs in the AC invasion GRN.** Fluorescent micrographs depicting BM (*lam::GFP*) and AC (*cdh-3p::mCherry::moeABD*) expression of endogenously tagged TFs of the cell cycle-dependent subcircuit (*egl-43::GFP::egl-43* (**A**), *GFP::hlh-2* (**B**), and *nhr-67::GFP* (**C**)) and cell cycle-independent subcircuit (*GFP::fos-1a* (**D**)) of the AC GRN in animals treated with empty vector control (left) or *swsn-4(RNAi)* (right). White arrowheads indicate ACs, yellow arrowheads indicate boundaries of breach in BM. Scale bar, 5μm. (**E**) Quantification of fluorescent expression of each TF::GFP in ACs of control animals and animals treated with *swsn-4(RNAi)*. Statistical comparisons were made between the expression of each TF subunit in the AC in control and RNAi-treated animals using Student's *t*-test (n≥30 for each condition; p values are displayed above black brackets). n.s. not significant.

## The PBAF assembly regulates AC contact with underlying BM

Previous investigations into SWI/SNF have demonstrated divergent roles for the PBAF assembly in cell cycle regulation. In yeast, Remodeling the Structure of Chromatin (RSC), the homologous complex to PBAF, is required for progression through mitosis [73, 74]. In *Drosophila*, the homologous complex PBAP does not appear to be required for mitotic progression; rather, cycling and $G_2/M$ transition is solely regulated by the BAF/BAP assembly [52]. In the *C. elegans* M lineage, RNAi-mediated loss of BAF subunits results in hyperproliferation of the developing tissue, whereas knockdown of PBAF subunits has little effect on cell cycle control [36]. Similarly, in this study, RNAi-mediated loss of PBAF subunits *pbrm-1*, *swsn-7*, or *swsn-9* resulted exclusively in single non-invasive cells expressing the AC reporter and with DHB::GFP C/N ratios indicative of G0 arrest (**Figs 2 and 5**). However, given that the enhanced *pbrm-1(RNAi)* resulted in much weaker endogenous protein knockdown than the enhanced RNAis targeting either the SWI/SNF ATPase (*swsn-4*) or BAF assembly subunit (*swsn-8*) in the AC (**S2C and S2D Fig**), and the dose-dependent phenotype following loss of the core ATPase (**Fig 4**), it is possible that we failed to observe a mitotic non-invasive AC phenotype due to insufficient PBAF subunit knockdown. To exclude this possibility, we next asked whether *strong* loss of PBAF subunit expression contributes to the mitotic non-invasive AC phenotype. To accomplish this, we used an auxin inducible degron (AID)-RNAi combination knockdown strategy [75, 76]. We generated a strain with *pbrm-1* endogenously labeled with mNeonGreen and an auxin inducible degron (AID) (*pbrm-1::mNG::AID*) in a genetic background containing AC (*cdh-3p::mCherry::moeABD*) and BM (*laminin::GFP*) reporters. We then quantified fluorescence expression in the AC in this strain. When grown under standard conditions, 6% of the ACs had not invaded the BM by the P6.p 4-cell stage, suggesting a partial loss of function of *pbrm-1* (n = 30) (**Fig 8A and 8G**). This partial loss of function phenotype is likely due to the insertion of the mNG::AID tag into the genomic locus, causing a putative hypomorphic allele. Next, we introduced a ubiquitous, mRuby-labeled TIR1 transgene (*eft-3p::TIR1::mRuby*) into the animals and assessed AC invasion under standard conditions (aux(-)) or in the presence of the auxin hormone (aux(+)) (**Fig 8B**).

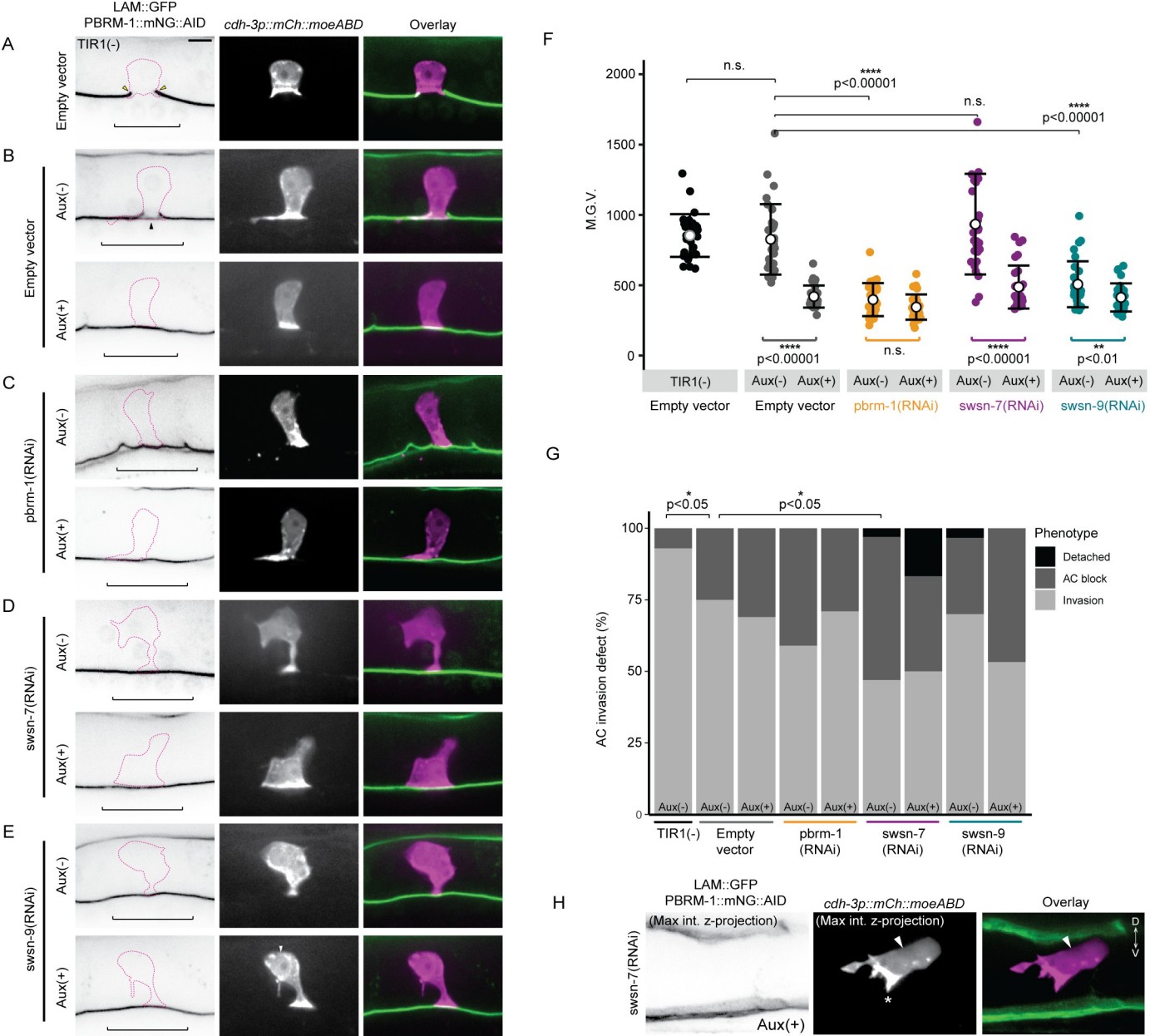

**Fig 8. PBAF promotes AC contact to the BM.** Representative micrographs of BM (*lam::GFP*) and endogenous *pbrm-1*::*mNG*::*AID* (left), AC (*cdh-3p*::*mCherry*::*moeABD*, center), and fluorescent overlays (right) of animals lacking **(A)** or possessing **(B-E)** ubiquitous TIR1 expression treated with empty vector control **(B)** or RNAi targeting PBAF subunits in the absence Aux(-) (top) or presence Aux(+) (bottom) **(C-E).** PBRM-1::mNG::AID images have been inverted for ease of visualization. Magenta dotted lines delineate boundaries of ACs. Scale bar, 5μm. **(F)** Quantification of fluorescence expression (M.G.V) of PBRM-1::mNG::AID in ACs of animals in each condition (N≥30 animals in each treatment; p values for Fisher's exact test comparing strains containing TIR1 to the TIR1(-) strain, and comparing strains containing TIR1 in the Aux(-) to the Aux(+) condition, are displayed above compared groups). **(G)** Stacked bar chart showing percentage of AC invasion defects corresponding to each treatment, binned by AC phenotype (N≥30 animals per condition; Fisher's exact test determined significance for penetrance of AC invasion defects between indicated conditions; all groups were compared and only significant comparisons were displayed). Black brackets indicate statistical significance between total invasion defect in each condition. **(H)** Max intensity z-projection of AC and BM reporter channels depicting a detached AC phenotype in *swsn-7*-deficient AC in the Aux(+) condition. BM (left), AC (center), fluorescence overlay (right). Asterisk in middle panel indicates polarized F-actin driven protrusion extending ventrally.

We observed no statistically significant difference in the fluorescence expression of PBRM-1::mNG::AID protein in the AC, nor did we observe any differences in AC invasion defects between the strain lacking the TIR1 transgene and the strain containing TIR1 grown on aux(-) media (TIR1+: 3% depletion, 17% invasion defect; n = 30; **Fig 8A, 8B and 8F**). However, in both conditions, some ACs that invaded seemed to do so only partially, as we noted cases where animals were lacking gonadal BM beneath the AC, but the ventral epidermal BM remained intact (**Fig 8B**, black arrowhead). This unique partial invasion phenotype appears specific to PBAF, as we failed to observe instances where only one BM (epidermal) remained at the P6.p 4-cell stage across all other treatments including BAF RNAi treatment.

In the aux(+) condition, there was a significant reduction in PBRM-1::mNG::AID protein level in the AC of animals containing the TIR1 transgene relative to the same strain grown in the aux(-) condition or the strain without the TIR1 transgene (49% and 51% depletion, respectively; n = 30) (**Fig 8B and 8F**). This result suggests that the auxin inducible degron in the *pbrm-1::mNG::AID* strain remains sensitive to TIR1-mediated degradation. Despite significant reduction in PBRM-1 expression, there was no significant difference in the penetrance of AC invasion defects in animals treated with auxin (17% invasion defect; n = 30) (**Fig 8G**). Like our previous results with *pbrm-1(RNAi)* treated animals, we observed no extra cells expressing the AC reporter following loss of expression of PBAF in the AC using the AID system.

Next, we treated *pbrm-1::mNG::AID* animals containing ubiquitous TIR1 with *pbrm-1 (RNAi)* in both aux(-) and aux(+) conditions. As expected, treatment of the *pbrm-1::mNG:: AID* strain with *pbrm-1(RNAi)* resulted in very low expression of the subunit in the AC even in the absence of auxin and there was no significant difference in expression between the Aux (-) and Aux(+) conditions (**Fig 8C and 8F**). Interestingly, there was also no significant difference in the penetrance of AC invasion defects between the *pbrm-1::mNG::AID* strain treated with control compared to the strain treated with *pbrm-1(RNAi)* in the presence of auxin (**Fig 8G**). Since the combination treatment of a hypomorphic *pbrm-1* allele, Auxin-AID-mediated depletion of endogenous PBRM-1::mNG::AID, and *pbrm-1(RNAi)* does not result in a significant increase in AC invasion defects or non-invasive mitotic ACs, these results suggest that, unlike the dose-dependent contribution to invasion of *swsn-4*, the *pbrm-1* strong knockdown or null phenotype may be only partial/incomplete loss of AC invasion.

Since the PBAF assembly in *C. elegans* consists of several subunits, *pbrm-1* (PBRM1), *swsn-7* (ARID2), and *swsn-9* (BRD7/BRD9), we next investigated whether combinatorial knockdown of PBAF subunits would enhance the penetrance of AC invasion defects or result in the mitotic non-invasive AC phenotype. In the absence of auxin, there was no significant difference in PBRM-1::mNG::AID expression in the AC of animals treated with *swsn-7(RNAi)* compared to animals treated with empty vector control (n = 30) (**Fig 8D and 8F**), however there was a significant increase in loss of AC invasion (50% invasion defect; n = 30) (**Fig 8G**). Strikingly, in one case, the AC was completely detached from the BM, as we detected no AC membrane protrusions (*cdh-3p::mCherry::moeABD*) in contact with the ventral surface of the gonad (**Fig 8H**). Animals treated with *swsn-7(RNAi)* and aux(+) had significantly lower expression of PBRM-1::mNG::AID in the AC when compared to animals treated with *swsn-7(RNAi)* in the aux(-) condition (49% depletion; n = 30) (**Fig 8F**). While no significant difference was seen in loss of AC invasion in aux(+) (48% AC invasion defect), 16% (5/31) of animals in this treatment had ACs entirely detached from the ventral BM (n = 31) (**Fig 8G and 8H**). In contrast to treatment with *swsn-7(RNAi)*, in the *swsn-9(RNAi)* aux(-) condition, PBRM-1::mNG::AID expression in ACs was significantly lower than that in the ACs of animals treated with empty vector control aux(-) (39% depletion; n = 30) (**Fig 8F**). It is unclear why transcriptional knockdown of *swsn-9* specifically results in a decrease in PBRM-1 protein expression in the AC and we theorize this may be the result of a potential stabilizing interaction between the SWSN-9

and PBRM-1 proteins. Despite this, we did detect a further decrease in the expression of PBRM-1::mNG::AID in ACs in *swsn-9(RNAi)* aux(+) compared to the *swsn-9(RNAi)* aux(-) condition (19% depletion; n = 30) (**Fig 8F**), however, we saw no statistically significant difference in penetrance of AC invasion defects between the two conditions (30% vs. 43%; n = 30) (**Fig 8G**). We also noted one animal with a detached AC in the *swsn-9(RNAi)* aux (-) condition and zero in the aux(+) condition (**Fig 8G**). Importantly, we only observed one AC per animal across all combinatorial treatments, supporting the hypothesis that the PBAF assembly does not contribute to $G_0$ cell cycle arrest in the AC.

Detached ACs in both the *swsn-7(RNAi)* and *swsn-9(RNAi)* AID combination knockdown conditions suggest that the PBAF assembly regulates AC contact with the ventral epidermal BM. A previous study has shown that AC-BM attachment is regulated by the *fos-1/egl-43* cell cycle-independent subcircuit of the AC GRN via regulation of lamellipodin/*mig-10b* and non-autonomously via netrin/*unc-6* signaling [77]. ACs deficient in components of this pathway are attached to the ventral epidermal BM when specified and gradually lose contact over time, with peak loss of contact occurring at the time of AC invasion at the P6.p 4-cell stage [77]. To determine whether the PBAF assembly remodels chromatin to promote activation of this subcircuit of the AC GRN, we treated endogenously tagged *fos-1::GFP* [22] animals with *pbrm-1 (RNAi)* and quantified fluorescence expression in ACs that displayed invasion defects (**S6A and S6B Fig**). Animals treated with *pbrm-1(RNAi)* had a modest but significant loss of FOS-1:: GFP protein levels in non-invasive ACs (34% depletion; n = 20) (**S6B Fig**), suggesting that the PBAF assembly partially regulates the *fos*-dependent pathway that mediates attachment to the underlying BM.

Since depletion of the PBAF assembly resulted in moderate loss of FOS-1::GFP in the AC, we next examined functional interactions between FOS-1 and PBRM-1. Given that the PBRM-1::mNG::AID allele was slightly hypomorphic, with ~17% invasion defects in backgrounds with TIR1, we used the strain containing TIR1 as a sensitized background. We found that even without the addition of auxin, co-depletion with *fos-1(RNAi)* resulted in almost complete loss of AC invasion (97% invasion defect; n = 31) (**S6C and S6D Fig**). Finally, we examined whether RNAi-mediated depletion of *pbrm-1* is synergistic with loss of downstream targets of FOS-1, the matrix metalloproteinases (MMPs). Previously, it has been shown that animals harboring null mutations for five of the six MMPs encoded in the *C. elegans* genome (*zmp-1,-3,-4,-5* and -6), show delayed AC invasion [21]. RNAi depletion of *pbrm-1* in quintuple MMP mutants significantly and synergistically enhanced late invasion defects (scored at the P6.p 8-cell stage) in this background (24% invasion defect; n = 33) (**S6E and S6F Fig**) as compared to loss of either *pbrm-1* (4%; n = 52) or MMPs (0%; n = 35) alone. Together, these results suggest that the PBAF assembly functions synergistically with FOS-1 to regulate AC invasion.

## Discussion

Previous work in the *C. elegans* AC and in cancer cell invasion has emphasized the necessity for dynamic chromatin states and chromatin regulating factors in the promotion of cellular invasion [24, 78–82]. In this study, we used the *C. elegans* AC as a single cell, *in vivo* system to identify a suite of chromatin factors that contribute to the process of cellular invasion. We performed a tissue-specific RNAi feeding screen to assess 269 genes implicated in chromatin binding, chromatin remodeling complexes, or histone modification. We do not claim that genes which we failed to identify as regulators of cellular invasion in the screen are unimportant for the process; however, RNAi-mediated loss of most chromatin factors in the screen did result in some penetrance of AC invasion defects (**S1 Table**). This finding was expected, as many of the genes we screened are global regulators of the genome and broadly contribute to

various cell biological processes including housekeeping and general maintenance. We extracted a list of the most penetrant regulators of cell invasion from the broader list (S2 Table). Many genes and gene classes that we recovered as significant regulators of AC invasion are homologous to human genes that have been previously studied in the context of cellular invasion and tumorigenesis including *cec-6*/CBX1/CBX8 [81, 83], *cfi-1*/ARID3A/ARID3C [84], *psr-1*/JMJD6 [85], *skp-1*/SNW1 [86], and several TAFs (*taf-1*/TAF1/TAF1L, taf-5/TAF5/TAF5L, *taf-7.1*/ TAF7/TAF7L) [87–89]. Additionally, we recovered nematode-specific genes including *nra-3*, *and cec-2*, and genes whose human homologs have not been previously studied in the context of cellular invasion to our knowledge, such as *cec-3* (homologous human protein is uncharacterized) and *gna-2*/GNPNAT. Interestingly, the roster of 'hits' from our RNAi screen includes both genes predicted to have activating (e.g. *mrg-1*/MORF4L1,2 and *mys-2*/KAT8) as well as repressive (e.g. *set-9*/SETD5 and *unc-37*/TLE3) transcriptional roles. Since the majority of the genes we identified as significant regulators of AC invasion have been previously studied in the context of invasion in human development and cancer metastasis, these results demonstrate the utility of the *C. elegans* AC invasion system as a genetically and optically tractable *in vivo* environment to corroborate and characterize previously identified chromatin factors that promote cellular invasion in human diseases such as rheumatoid arthritis and cancer.

For the majority of this study, we focused on characterizing the contribution of the SWI/ SNF ATP-dependent chromatin remodeling complex to cellular invasion as it was highly represented among our list of significant regulators of AC invasion (S2 Table) and has been extensively studied in the context of both cellular invasion and cell cycle control across a variety of animal models and in human cancers [32, 36, 41, 43, 49, 56, 58, 59, 82, 90–94]. Prior whole-exome studies have determined that over 20% of human tumors harbor mutations in one or more subunits of the SWI/SNF complex [33, 49, 95]. Among the most frequently mutated subunits of the chromatin remodeling complex throughout SWI/SNF-deficient cancers is the core ATPase subunit BRG1/SMARCA4 [49, 96] and the mutually exclusive ATPase paralog to BRG [33, 80, 95, 97]. Previous investigation has determined BRM to be an effective synthetic lethal target in BRG1-deficient cancer, and vice-versa [98, 99]. Despite the compensatory nature of BRG1/BRM in many tumorigenic contexts, concomitant loss of expression of the ATPases has been described in metastatic murine models and patient-derived non-small-cell lung cancer (NSCLC) cell lines and is associated with poor patient survival [90, 100, 101]. In *C elegans*, the sole SWI/SNF ATPase, *swsn-4*, has a high degree of homology to both mammalian BRG1 and BRM, providing a unique opportunity to accessibly model the connection between the dual loss of BRG1/BRM associated with poor prognostic outcomes in NSCLC and cellular invasion in the AC.

Here we used the *C. elegans* AC invasion system as a model to investigate whether the dose-dependent relationship between the SWI/SNF ATPase and differentiated phenotype extends to cellular invasion. To this end, we report enhancement of all endogenously tagged subunits of the complex in the AC relative to neighboring VU and VPC tissues (S4 Fig). While it is tempting to interpret the enhancement of SWI/SNF subunit expression in the AC as evidence for the dependence of cellular invasion on SWI/SNF activity, it is also possible that this difference in expression is a consequence of terminal differentiation, since at the time of invasion, the AC is terminally differentiated (unlike the VU). In line with this argument, from the P6.p 1-cell stage to the 4-cell stage SWSN-4, SWSN-8, and PBRM-1 protein levels are lower in the VPCs relative to their expression in the AC, however expression of all three SWI/SNF subunits rises in the VPCs at the 8-cell stage when the primary vulva cells have terminally differentiated (Fig 3C', 3D' and 3E').

By assessing AC invasion phenotypes at wildtype levels of SWSN-4 and in moderate and severe ATPase knockdown conditions (Fig 4), we find that cellular invasion and cell cycle

control depends on the dose of functional SWI/SNF present in the AC. Generally, enhancement of the AC mitotic phenotype statistically tracked with a progressive step down in mean expression of the ATPase in the AC across our experiments. Further analysis of these SWI/SNF::GFP strains suggested that intra-complex and inter-assembly regulation exists in the AC at the time of invasion with both SWI/SNF assemblies cooperating to activate expression of the ATPase (**S3D Fig**). It is possible that this added level of complex autoregulation contributes to an "optimal" dose of the ATPase in a cell- and context-specific manner.

In addition to reflecting the dose-dependent nature of the SWI/SNF ATPase in cancer, our data in the AC is consistent with work done in *C. elegans* early mesoblast development where complete loss of the *swsn-4* ATPase using a catalytically dead mutant and lineage-specific knockout strategy results in loss of cell cycle arrest [36]. Although we cannot be sure that combining *swsn-4(RNAi)* with an antiGFP-targeting nanobody to deplete the SWI/SNF ATPase results in complete loss of protein expression, we show that treatment with the improved *swsn-4(RNAi)* vector alone is sufficient to phenocopy the null phenotype previously reported in late mesoblast (SM) development (**S4 Fig**). Altogether, this data supports the hypothesis that SWI/SNF cell-autonomously contributes to cell cycle control in a dose-dependent manner and provides the first line of evidence to link SWI/SNF ATPase dosage to the dichotomy between invasion and proliferation (**Fig 9**).

While previous work in our lab, based on localization of a DNA licensing factor, CDT-1, has demonstrated indirectly that ACs must arrest in a $G_0/G_1$ cell cycle state [22, 24], we lacked a sensitive enough tool to distinguish between these two interphase states. From our recent work utilizing a CDK sensor to examine the proliferation-quiescence decision in *C. elegans*, we can distinguish between pre-terminal cells in the somatic gonad in $G_1$ (mean C/N ratio: 0.67 +/-0.10) and terminally differentiated $G_0$ uterine cells (mean C/N ratio: 0.30+/-0.11) [37]. Here, we compare CDK activity measurements in the ACs of control animals with that of the terminal Pi lineage to provide the first quantitative demonstration that ACs arrest in a CDK$^{low}$ $G_0$ state to invade (**Fig 5A, 5E and 5F**). Furthermore, by combining the CDK sensor with loss of SWI/SNF subunits, our data indicate that the SWI/SNF BAF assembly is specifically responsible for regulation of $G_0$ cell cycle arrest in the AC. Here, using a CDK sensor [37, 70], we show that loss of either core or BAF assembly subunits specifically results in mitotic ACs that failed to invade the BM. Our cell cycle sensor data establishes that a major contribution of the BAF assembly to AC invasion is through maintenance of $G_0$ arrest, as many ACs that failed to invade the BM had increasing CDK activity, indicative of cells cycling in $G_1$, S or $G_2$. Alternatively, 14% of ACs that failed to invade the BM following loss of *swsn-4*/ATPase of the complex had CDK activity ratios indicative of $G_0$ cell cycle arrest, suggesting a cell cycle-independent defect. In support of this, forced $G_0$ arrest of BAF-deficient ACs was sufficient to significantly rescue invasion, whereas CKI-1 induction failed to rescue invasion in ACs with RNAi-mediated loss of *swsn-4*/ATPase. Altogether, our results indicate that the SWI/SNF complex contributes to AC invasion through regulation of $G_0$ cell cycle arrest via the BAF assembly. Further investigation will require biochemical techniques to identify cell cycle regulators and TF targets of the BAF assembly to provide a mechanistic explanation for how exactly BAF regulates the chromatin landscape to promote invasion (**Fig 9**, blue arrow). Targeted DNA adenine methyltransferase identification (TaDa) is an attractive biochemical approach that may be adaptable to the AC invasion system, as this approach has been characterized as an effective, tissue-specific method to identify TF-target sequence interactions in the *C. elegans* epidermis [102].

Previous work in *C. elegans* has not revealed a connection between the PBAF assembly and cell cycle arrest. Our initial experiments with improved RNAi vectors targeting PBAF subunits resulted in a lower penetrance of AC invasion defects relative to loss of core or BAF subunits.

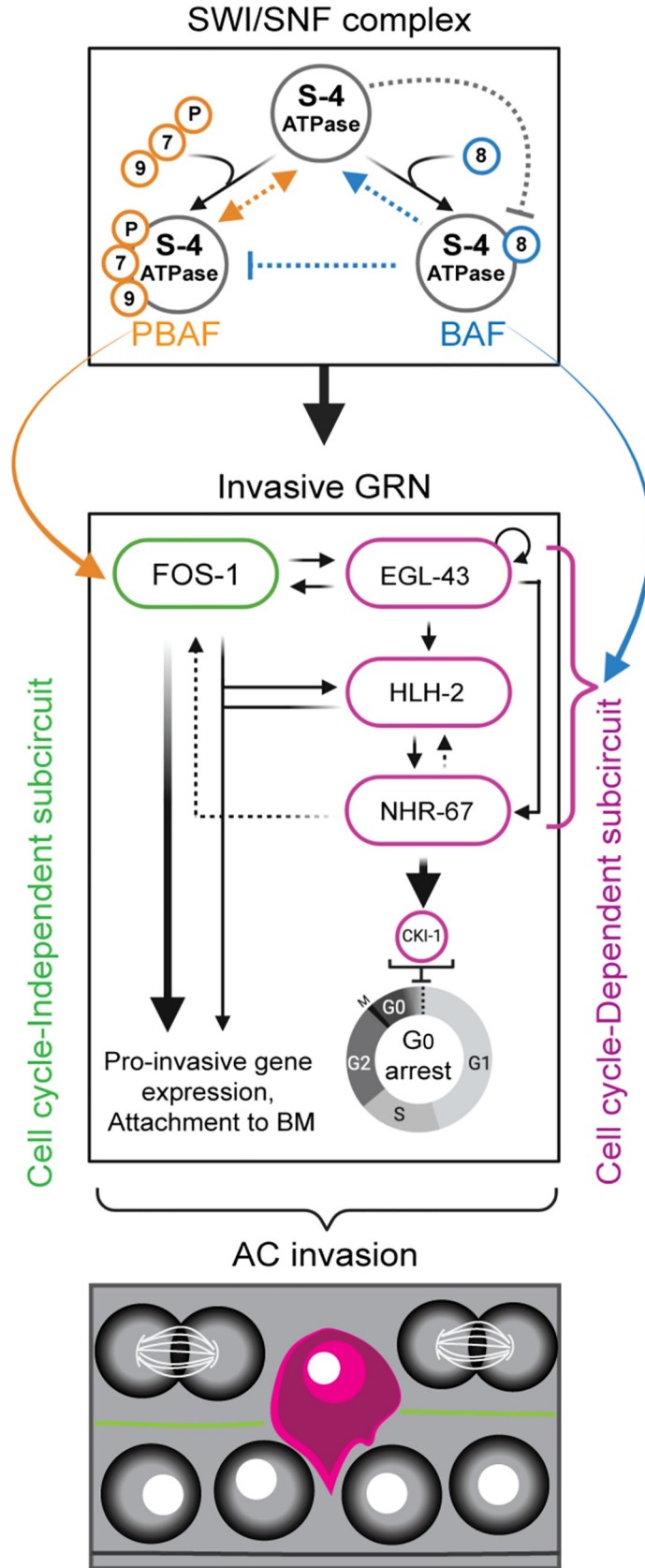

**Fig 9. SWI/SNF complex assemblies promote AC invasion.** Schematic summary of the how the SWI/SNF ATPase (S-4, *swsn-4*), PBAF (orange–S-7, swsn-*7*; S-9, *swsn-9*; P, *pbrm-1*), and BAF (blue–S-8, *swsn-8*) assemblies contribute to AC invasion at the distinct levels of pro-invasive gene expression and BM attachment (left, green) and cell cycle arrest (right, magenta).

Additionally, our CDK sensor data suggested that non-invasive ACs deficient in *pbrm-1* remain in $G_0$. Thus, our data shows no PBAF contribution to cell cycle control in the AC. To confirm this, we used the auxin inducible degron (AID) system to robustly deplete the PBAF assembly through combined loss of endogenous PBRM-1::mNG::AID with RNAi-mediated knockdown of either of the other two PBAF assembly subunits, *swsn-7* or *swsn-9*. This combination knockdown strategy corroborated our previous results as we saw no significant penetrance of extra ACs. Rather, here we associate a striking AC detachment phenotype with strong combined knockdown of the PBAF assembly subunits. We also note aberrant BM morphology in some ACs deficient in PBAF subunits, with only one of the two BMs removed, suggesting that this assembly regulates attachment and extracellular matrix (ECM) remodeling in wild-type ACs to promote invasion. We hypothesize that the PBAF assembly is regulating ventral BM attachment and ECM remodeling potentially through the regulation of HIM-4/Hemicentin, an extracellular immunoglobulin-like matrix protein that functions in the AC to fuse the two BMs through the formation of a novel BM-BM adhesion, the B-LINK [103]. Finally, although RNAi-mediated transcriptional knockdown of PBAF assembly subunits only partially depleted levels of FOS-1::GFP, a key TF responsible for the expression of MMPs and other pro-invasive targets, we detected significant enhancement of invasion defects when depleting *fos-1* in a putative hypomorphic *pbrm-1* background. Reciprocally, depletion of *pbrm-1* enhanced the invasion defect of a quintuple MMP mutant. Since we noted multiple instances of AC-BM detachment following PBAF assembly subunit depletion, we propose that PBAF functions in part with FOS-1 to facilitating activating chromatin states at the regulatory regions of pro-invasive genes required for BM attachment.

To conclude, we emphasize that the primary insight of the data presented here should be assessed with respect to the varied, pleiotropic effects the SWI/SNF complex has in the regulation of dynamic differentiation and cellular behaviors across *C. elegans* development. Previous studies have characterized the effects of the loss of specific SWI/SNF subunits in many cell types in the developing animal, such as in the context of hermaphrodite-specific neuron (HSN) migration and serotonin expression [59]. The current study corroborates an emerging theme in the investigation of the *C. elegans* SWI/SNF complex which is, in general, compromising the function or functional dose of SWI/SNF subunits in *C. elegans* effects dynamic cell behaviors. In addition to corroborating a general role for SWI/SNF across nematode development, this investigation into the role of the complex in the promotion of AC invasion reveals the distinct contribution of each SWI/SNF assembly to the process of cellular invasion at the phenotypic level, provides evidence for single-cell SWI/SNF assembly-specific invasive mechanisms, and establishes a visually tractable platform with which to investigate the conserved requirement for SWI/SNF and other chromatin factors in cellular invasion.

## Materials & methods

### *C. elegans* strains and culture conditions

All animals were maintained under standard conditions and cultured at 20˚-25˚C, except strains containing temperature-sensitive alleles *swsn-1(os22)*, *swsn-4(os13)*, and the uterine-specific RNAi hypersensitive strain used in the chromatin remodeler screen containing the *rrf-3(pk1426)* allele, which were maintained at either 15˚C or 20˚C [104]. The heat shock

inducible *cki-1*::*mTagBFP2* transgene was expressed via incubating plates of animals at 32˚C for 2–3 hours in a water bath starting at the P6.p 2-cell VPC stage. Animals were synchronized for experiments through alkaline hypochlorite treatment of gravid adults to isolate eggs [105]. In the text and figures, we designate linkage to a promoter through the use of a (p) and fusion of a proteins via a (::) annotation.

## Molecular biology and microinjection

SWI/SNF subunits *swsn-4* and *swsn-8* were tagged at their respective endogenous loci using CRISPR/Cas9 genome editing via microinjection into the early adult hermaphrodite syncytial gonad [63, 106]. Repair templates were generated as synthetic DNAs from either Integrated DNA Technologies (IDT) as gene blocks (gBlocks) or Twist Biosciences as DNA fragments and cloned into *ccdB* compatible sites in pDD282 by New England Biolabs Gibson assembly [107]. Homology arms ranged from 690–1200 bp (see S5 **Table** for additional details). sgRNAs were constructed by EcoRV and NheI digestion of the plasmid pDD122. A 230 bp amplicon was generated replacing the sgRNA targeting sequence from pDD122 with a new sgRNA and NEB Gibson assembly was used to generate new sgRNA plasmids (see **S5 Table** for additional details). Hermaphrodite adults were co-injected with guide plasmid (50 ng/μL), repair plasmid (50 ng/μL), and an extrachromosomal array marker (pCFJ90, 2.5 ng/μL), and incubated at 25˚C for several days before screening and floxing protocols associated with the SEC system [107].

## RNA interference (RNAi)

All 269 RNAi clones assessed in the chromatin remodeler screen were derived from the commercially available Vidal or Ahringer RNAi libraries. Presence of inserts into the L4440 RNAi vector was confirmed via colony PCR amplification of all L4440 vectors used in the chromatin remodeler screen. Vectors which resulted in penetrant loss of invasion (see S2 **Table**) were also sequenced to confirm the identity of the insert targeting chromatin remodeler genes in the L4440 vector using Sanger sequencing at the Genomics Core Facility at Stony Brook University. An RNAi sub-library of SWI/SNF subunits was constructed by cloning 950–1000 bp of synthetic DNA based on cDNA sequences available on WormBase (www.wormbase.org) into the highly efficient T444T RNAi vector [108, 109] (see **S3 Table**). Synthetic DNAs were generated by Twist Biosciences as DNA fragments and cloned into restriction digested T444T using NEB Gibson Assembly. For all experiments, synchronized L1 stage animals were directly exposed to RNAi through feeding with bacteria expressing dsRNA [110].

## Auxin-mediated degradation

To combine RNAi with the depletion of AID-tagged proteins, 1 mM K-NAA was used, and its effects were analyzed as previously described [111]. Briefly, L1 animals were first synchronized by sodium hypochlorite treatment and transferred to NGM plates seeded with the RNAi vector of interest. At the P6.p 1-cell stage, a time in development where the AC has already undergone specification, animals were transferred to RNAi-seeded plates treated with K-NAA. Animals were staged by DIC.

## Live cell microscopy

All micrographs included in this manuscript were collected on a Hamamatsu Orca EM-CCD camera mounted on an upright Zeiss AxioImager A2 with a Borealis-modified CSU10 Yoka-gawa spinning disk scan head using 405nm, 488 nm, and 561 nm Vortran lasers in a VersaLase

merge and a Plan-Apochromat 100x/1.4 (NA) Oil DIC objective. MetaMorph software (Molecular Devices) was used for microscopy automation. Several experiments and all RNAi screening were scored using epifluorescence visualized on a Zeiss Axiocam MRM camera, also mounted on an upright Zeiss AxioImager A2 and a Plan-Apochromat 100x/1.4 (NA) Oil DIC objective. Animals were mounted into a drop of M9 on a 5% Noble agar pad containing approximately 10 mM sodium azide anesthetic and topped with a coverslip.

### Assessment of AC invasion

Both for the purposes of the chromatin factor RNAi screen and most other experiments, AC invasion was scored at the P6.p 4-cell stage, when 100% of wild-type animals exhibit a breach in the BM [14]. AC invasion was scored at the P6.p 8-cell stage for the purposes of assessing invasion delay enhancement of the quintuple MMP mutant when treated with *pbrm-1(RNAi)* (**S6E and S6F Fig**). In strains with the laminin::GFP transgene, an intact green fluorescent barrier under the AC was used to assess invasion. Wild-type invasion is defined as a breach as wide as the basolateral surface of the AC [14]. Raw scoring data is available in **S1 and S4 Tables.**

### Image quantification and statistical analyses

Images were processed using Fiji/ImageJ (v.2.1.0/1.53c) [112]. Expression levels of GFP:: SWSN-4, SWSN-8::GFP, PBRM-1::eGFP, and PBRM-1::mNG::AID were measured by quantifying the mean gray value of AC nuclei, defined as somatic gonad cells near the primary vulva expressing the *cdh-3p*::*mCherry*::*moeABD* transgene. Background subtraction was performed by rolling ball background subtraction (size = 50). For characterization of experiments involving SWI/SNF endogenous tags and AC GRN TFs::GFP treated with *SWI/SNF(RNAi)* and GFP-targeting nanobody the L3 stage, only animals exhibiting defects in invasion were included in the analysis. Data was normalized to negative control (empty vector) values for the plots in **Figs 4** and **S6**. Quantification of either CDK cell cycle sensor (either DHB::GFP or DHB::2xmKate2) was performed by hand, as previously described [37]. Images were overlaid and figures were assembled using Adobe Photoshop 2020 (v. 21.1.2) and Adobe Illustrator 2020 (v. 24.1.2), respectively. Statistical analyses and plotting of data were conducted using RStudio (v. 1.2.1335). Statistical significance was determined using either a two-tailed Student's t-test or Fisher's exact probability test. Figure legends specify when each test was used and the p-value cut-off.

### Supporting information

**S1 Fig. AC invasion is disrupted in temperature sensitive SWI/SNF hypomorphs.** Single planes of confocal z-stacks representing AC invasion in *swsn-1(os22) and swsn-4(os13)* temperature sensitive mutants with fluorescently labeled AC (magenta, *cdh-3>mCherry*::*moeABD*) and BM (green, *laminin*::*GFP*) scored at the permissive temperature (**A**) and restrictive temperature (**B**). Significant loss of invasion was seen in both *swsn-1(os22)* (20% loss of invasion) and *swsn-4(os13)* (24% loss of invasion) hypomorphic[ts] strains when grown at the restrictive temperature 25˚C and assessed at the P6.p 4-cell 1˚ VPC stage (**B**). White arrowheads indicate ACs. Yellow arrowheads in A indicate boundaries of breaches in the BM. Numbers in bottom right of fluorescence overlay panel in A indicate penetrance of wildtype AC invasion. Numbers in bottom right of fluorescence overlay panel in B indicate penetrance of invasion defects. (PDF)

**S2 Fig. Improved SWI/SNF RNAi significantly knocks down SWI/SNF expression in the AC.** Fluorescent micrographs depicting BM (*laminin*::*GFP*) and expression of SWSN-4::GFP **(A)**, SWSN-8::GFP **(B)**, and PBRM-1::eGFP **(C)** in the AC in animals fed empty vector control (left) or RNAi targeting the endogenous allele (right). White arrowheads indicate ACs, yellow arrowheads indicate boundaries of breach in BM, and white brackets indicate 1 VPCs. Scale bar, 5μm. **(D)** Corresponding quantifications of fluorescent expression. Statistical comparisons were made between the expression of each SWI/SNF subunit in the AC in control and RNAi-treated animals using Student's *t*-test (n≥30 for each stage and subunit; p values are displayed above compared data). **(E)** Stacked bar chart showing percentage of AC invasion defects corresponding to each treatment, binned by AC phenotype (n≥30 animals per condition).
(PDF)

**S3 Fig. SWI/SNF subunits exhibit intra-complex and inter-assembly regulation. (A-C)** Representative fluorescence micrographs depicting endogenous GFP expression of individual SWI/SNF subunits representative of the core (*swsn-4*, **A**), BAF assembly (*swsn-8*, **B**), and PBAF assembly (*pbrm-1*, **C**) in the AC (*cdh-3p*::*mCherry*::moeABD) following treatment with RNAi targeting each SWI/SNF assembly **(A),** or the core ATPase and alternative SWI/SNF assembly **(B-C)**. White arrowheads indicate ACs, yellow arrowheads indicate boundaries of breach in BM, and white brackets indicate 1 VPCs. Scale bar, 5μm. **(D)** Quantification of fluorescence expression (mean gray value) of endogenous subunits in each condition. Data is normalized to empty vector control across each strain. Statistical comparisons were made between the expression of each SWI/SNF subunit in the AC in control and RNAi-treated animals using Student's *t*-test (n≥30 for each stage and subunit; p values are displayed above compared data). n.s. not significant. **(E)** Quantification of fluorescence expression of endogenous GFP-tagged subunits of non-invasive ACs following loss of expression of alternative SWI/SNF subunits, binned per RNAi treatment by phenotype into single non-invasive AC (1AC) and mitotic non-invasive AC (2+ AC). Statistical comparisons (Student's *t*-test; p values are displayed above compared data) were limited to conditions with n>10 ACs in each phenotype. n.s. not significant. **(F)** Schematic summary of SWI/SNF core and assembly auto and cross regulation.
(PDF)

**S4 Fig. Improved *swsn-4* RNAi recapitulates SWI/SNF ATPase null phenotype in the sex myoblasts. (A)** Single confocal z-planes depicting DIC (left) and expression of lineage-restricted CDK sensor (*unc-62>DHB*::*2xmKate2*, right) in the vulva and SM cells at the P6.p 8-cell stage corresponding to the stage when wild-type SM cells differentiate and exit the cell cycle. Animals were treated with empty vector control (top) or *swsn-4(RNAi)* (bottom). All representative images in each treatment are derived from the same z-stack from the same animal in the corresponding z-plane (top-left). Average or individual C/N CDK sensor ratios are listed in the bottom-right of corresponding panels. White arrowheads indicate individual SM cells. White brackets indicate 1° VPCs. **(B)** Quantification of the number of SM cells present at the P6.p 8-cell stage in control and *swsn-4(RNAi)* treated animals. **(C)** C/N CDK sensor ratios for SM cells in each treatment. Gradient scale depicts cell cycle state as determined by quantification of each AC in all treatments (n≥30 animals per treatment), with dark-black depicting differentiation into $G_0/G_1$ and lighter-magenta depicting $G_2$ cell cycle states.
(PDF)

**S5 Fig. AC-specific expression of CKI-1 rescues invasion in BAF-depleted ACs. (A)** DIC (left) and fluorescent (right) images depicting BM (*laminin*::*GFP*) and AC-specific CKI-1 (*cdh-3>CKI-1*::*GFP*) in empty vector control animal (top) and animals treated with *swsn-4*

*(RNAi)* (middle) or *swsn-8(RNAi)* (bottom). Scale bar, 5μm. **(B)** Stacked bar chart showing quantification of percentage of AC invasion defects corresponding to each treatment (n≥30 animals per condition, p values for Fisher's exact test comparing invasion penetrance in control animals and animals with the rescue transgene (+CKI-1::GFP) are displayed above black brackets).
(PDF)

**S6 Fig. PBAF partially regulates the FOS-1 transcription factor. (A)** Representative DIC (left) and fluorescent (right) micrographs depicting expression of endogenous GFP::FOS-1a and BM (*laminin*::*GFP*) in control (top) and *pbrm-1(RNAi)* treated (bottom) animals. White arrowhead indicates ACs, yellow arrowheads indicate boundaries of the breach in the BM. Scale bar, 5μm. **(B)** Quantification of GFP::FOS-1a expression in ACs of control and *pbrm-1 (RNAi)* treated animals, normalized to mean expression of control group. Statistical comparisons were made between expression in the AC in control and RNAi-treated animals using Student's *t*-test (n≥20 for each condition; p value is displayed above black bracket). **(C)** DIC-Fluorescence overlay (left), and PBRM-1::mNG::AID and BM (LAM::GFP) (right), in animals treated with empty vector control (top) or *fos-1(RNAi)* (bottom). **(D)** Stacked bar chart showing percentage of AC invasion defects corresponding to each treatment and genetic background in C (n≥30 animals per condition, p values for Fisher's exact test comparing invasion defect penetrance in wild-type animals treated with *fos-1(RNAi)* and *pbrm-1::mNG::AID* animals treated with *fos-1(RNAi)* is displayed above black bracket). **(E)** Representative DIC (top-left), BM (LAM::GFP, top-right), AC (*cdh-3>PH*, bottom-left), and overlay (bottom-right) of P6.p 8-cell vulva in an MMP-deficient (-) animal treated with *pbrm-1(RNAi)*. **(F)** Stacked bar chart showing percentage of AC invasion defects corresponding to each treatment and genetic background in E (n≥30 animals per condition, p values for Fisher's exact test comparing invasion defect penetrance in wild-type animals treated with *pbrm-1(RNAi)* and MMP (-) animals treated with *pbrm-1(RNAi)* is displayed above black bracket).
(PDF)

**S1 Table. Chromatin factors assessed for AC invasion contribution (see excel file).** 270 chromatin regulating factors targeted by RNAi for AC invasion defects. n≥30 animals for each RNAi clone. For each RNAi clone tested, the corresponding genetic sequence name, public name, protein annotation, and human homolog (HUGO Gene Nomenclature) from www.wormbase.com is given. Penetrance for each invasion defects is given as the % of animals with ACs that fail to invade the BM at the P6.p 4-cell stage out of the total number of animals assessed (Block/Invasion+Partial). Partial refers to cases where an animal had a breach in the BM narrower than the width of the basolateral surface of the invading AC. Genes in bold were recovered as significant regulators of AC invasion (S2 Table). Annotations were mined from the STRING consortium www.string-db.org. Asterisks in human ortholog column denote genes with > 5 detected human orthologs, for which only the first 5 returned orthologs were listed. N.A. denotes genes for which no human ortholog exists. List is organized alphabetically based on genetic sequence name.
(XLSX)

**S2 Table. Significant regulators of AC invasion (see excel file).** 41 chromatin and chromatin regulating factors (CRFs) identified as significant regulators of AC invasion. For each RNAi clone listed, the corresponding genetic sequence name, public name, and human homolog is listed. AC invasion scoring data is provided for each clone at the P6.p 4-cell stage. Genes were determined to be significant AC invasion regulators if RNAi targeting resulted in ≥ 20% loss of invasion at the P6.p 4-cell stage (n≥30 animals). Genes in bold are components of the SWI/

SNF complex. Asterisks denote genes previously published to regulate *C. elegans* AC invasion. N.A. denotes genes for which no human ortholog exists. List is organized alphabetically based on genetic sequence name.
(XLSX)

**S3 Table. Enhanced (T444T) RNAi vectors used in this study (see excel file).** Sequences correspond to the inserts cloned into novel enhanced RNAis in the T444T vector, which target 5 individual SWI/SNF subunits, and which were introduced in this study. *Fos-1*(RNAi) T444T clone was previously published, and the relevant sequence can be found in manuscript reference [22].
(XLSX)

**S4 Table. Strains used in this study (see excel file).** Information corresponding to each strain used in the study, including strain designation ('Strain' column), corresponding genotype ('Genotype' column), and practical description of the genotype is provided ('Description' column). Each strain is attributed to the figure(s) which contains data derived from the strain ('Figure(s)' column) and the relevant source of the strain is provided ('Source' column). Numbers in the 'Source' column for strains which were not originally generated in this study correspond to the manuscript reference number where the strain was first published.
(XLSX)

**S5 Table. CRISPR reagents (see excel file).** The relevant sgRNA and left/right homology arm sequences used to generate endogenous GFP (*swsn-4* and *swsn-8*) or mNG::AID (*pbrm-1*) knock ins in SWI/SNF subunits is provided, along with the terminus containing the insertion.
(XLSX)

**S1 Data. Quantification and statistical tests related to data (see archived excel files).**
(ZIP)

## Acknowledgments

We are thankful to David Gray, Ed Luk, Laura Mathies, Benjamin Martin, Robert Morabito, Valerie Reinke, Courtney Tello, and Gerald Thomsen for advice and comments on this manuscript. We would also like to thank Thom Geer of Nobska Imaging for advice and 'scientific enabling'. Some *C. elegans* strains were provided by the CGC, which is funded by NIH Office of Research Infrastructure Programs (P40 OD010440).

## Author Contributions

**Conceptualization:** Jayson J. Smith, David Q. Matus.

**Data curation:** Jayson J. Smith, Yutong Xiao, David Q. Matus.

**Formal analysis:** Jayson J. Smith, Yutong Xiao, Michael A. Q. Martinez, David Q. Matus.

**Funding acquisition:** David Q. Matus.

**Investigation:** Jayson J. Smith, Yutong Xiao, Nithin Parsan, Taylor N. Medwig-Kinney, Michael A. Q. Martinez, Frances E. Q. Moore, Nicholas J. Palmisano, Abraham Q. Kohrman, Mana Chandhok Delos Reyes, Rebecca C. Adikes, Simeiyun Liu, Sydney A. Bracht, Wan Zhang, David Q. Matus.

**Methodology:** Jayson J. Smith, David Q. Matus.

**Project administration:** David Q. Matus.

**Resources:** Kailong Wen, Paschalis Kratsios, David Q. Matus.

**Supervision:** David Q. Matus.

**Visualization:** Jayson J. Smith, David Q. Matus.

**Writing – original draft:** Jayson J. Smith.

**Writing – review & editing:** Jayson J. Smith, Yutong Xiao, Taylor N. Medwig-Kinney, Michael A. Q. Martinez, Frances E. Q. Moore, Nicholas J. Palmisano, Rebecca C. Adikes, Paschalis Kratsios, David Q. Matus.

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
