## [Decision Letter · Decision Letter 0]

10 Nov 2021

Dear Dr Matus,

Thank you very much for submitting your Research Article entitled 'The SWI/SNF chromatin remodeling assemblies BAF and PBAF differentially regulate cell cycle exit and cellular invasion in vivo' to PLOS Genetics.

The manuscript was fully evaluated at the editorial level and by three independent peer reviewers. The reviewers appreciated the attention to an important topic but identified some concerns that we ask you address in a revised manuscript

We therefore ask you to modify the manuscript according to the review recommendations, which were all focused on changes to the writing and data presentation in the manuscript. Your revisions should address the specific points made by each reviewer. 

[LINK]

Yours sincerely,

Jeremy Nance

Associate Editor

PLOS Genetics

Gregory P. Copenhaver

Editor-in-Chief

PLOS Genetics

Reviewer's Responses to Questions

**Comments to the Authors:**

Reviewer #1: This is a very interesting study describing the role of SWI/SNF complex in regulating cell cycle exit and invasiveness in the AC cell in C. elegans. The authors use many cutting edge techniques to tackle this question at the level of a single cell in a developing organisms. They show dose dependence of the effect and find differences between the roles of BAP and PBAF. I agree with the initial assessment that identifying the direct targets of BAF and PBAF would add a lot to the story, but I also agree with the authors that that it would not be feasible with currently available techniques. I also think that the experiments as presented advance the field sufficiently to warrant publication in PLoS Genetics. The authors already addressed a set of comments from previous reviewers, and added a fair amount of new data, which did improve the quality of manuscript. I have no major concerns.

I just have one minor comment. Maybe I missed it, but I could not find the in-text citation for the data regarding the role SWI/SNF in myogenesis in C. elegans, page 6 line 105-106 and page 7 line 107-111. Is there a reference missing? Or did I miss it?

Reviewer #2: This is an interesting (and quite substantial) study of the importance for the SWI/SNF complex in anchor cell invasion, a well-studied model for cell invasion across a basement membrane. It applies a number of cutting edge approaches to this problem including the use of inducible degrons to test for cell-autonomous effects, and the development of CRISPR knock-in alleles to assess gene expression. The authors should be commended for using their GFP CRISPR lines to measure the extent of RNAi-mediated knockdown, something that is unfortunately rarely done in C. elegans studies.

At a high level, the question of how we can understand the role of pleiotropic regulators like SWI/SNF in specific cells is an important one. While many of the results seem fairly “expected” from prior knowledge, this study does provide new advances. For example, the finding that genes that are part of specific SWI/SNF complexes BAF and PBAF give rise to apparently distinct phenotypes. The endogenously CRISPR-tagged alleles generated here will likely be a useful resource for other C. elegans labs.

The previous reviews for PLOS Biology raised several concerns, that in my view have been adequately addressed for publication in PLOS Genetics.

I have some concerns about how specific the results are to SWI/SNF and suggest that this be emphasized a bit more in the framing and interpretation of the paper. The RNAi screen produced a whopping 82 hits out of 269 “chromatin regulatory factors” screened. To me this suggests that almost any major global deficit in gene expression in the anchor cell is likely to lead to defects in invasion. Similarly, it seems likely, especially given the sex mesoblast results, that many/most other cells that undergo dynamic differentiation behaviors would be SWI/SNF dependent. I am curious what blocking RNA PolII e.g. with ama-1 RNAi would have in the anchor cell as sort of an “edge case.” Similarly, it is worth posing the question of whether similar SWI/SNF perturbations might similarly block the normal behaviors of other cells that are dynamic after embryogenesis such as Q cell migration, any dividing cells, Y to PDA, FLP/PVD arborization, vulval morphogenesis, etc. I’m hesitant to recommend doing new experiments to ask these questions as the paper already is bending under the weight of a huge amount of data, but they are worth discussing regardless. Obviously this discussion should still account for the fact that hits include both genes predicted to have activating and repressive functions, and the evidence that loss of BAF and PBAF component separately leads to defects in invasion in either cell cycle dependent or independent ways argues for some level of specificity.

Another key unanswered question is what molecular defects (presumably in anchor cell gene expression) underly the observed cell cycle and invasion phenotypes when each complex is perturbed. This is eminently addressable with modern techniques but in my view given the already large scope of this study is fine to leave for a future paper.

Minor points.

I would move figure S2/S3 to be a main figure

Figure 7G – are all the ‘NS’ bars necessary? Would be easier to follow if you just state in the legend that other comparisons are NS, highlighting the few significant comparisons.

Figure callouts for Figure 7 in the textare not right (for example Figure 7E for mNG quantification, 7F for “Detached” phenotype example

Reviewer #3: This manuscript by Smith et al uses the Anchor cell invasion in C elegans as a model system to identify new regulators of invasion. Using this elegant model system and genetic screens they find:

• SWI/SNF and its sub-complexes pBAF and BAF to control AC cell invasion.

• SWI/SNF acts in a dose dependent manner to regulate AC cell invasion.

• The PBAF and BAF control different aspects of invasion. While PBAF controls invasion via Fos and attachment to the basement membrane, BAF controls invasion via the cell cycle.

This manuscript has already undergone one round of review process and the authors have satisfactorily addressed the points brought up by the previous round of review. I agree with the authors that direct targets of these complexes are difficult to obtain currently due to limitations in current technology. In addition, I also feel that this manuscript has sufficient data to contribute significantly to the field.

My major issue is how the manuscript is written. As an outsider to the C elegans field, and contrary to the previous round of reviews, I thought it was a difficult manuscript read and navigate. Here are a few suggestions that the authors can use to simplify the manuscript so it can be accessible to the broader scientific community if they wish.

1. Use fewer acronyms. I feel like acronyms like “CRFs” are not required and one can spell this out. There are a lot of moving parts to this manuscript—using fewer acronyms will allow the reader to focus on the science.

2. While reading the initial part of the manuscript I was deeply worried about RNAi levels and phenotypic analysis. This was addressed in the latter part of the manuscript by using nulls – can they indicate to readers that this is in the offing? In my opinion, it will be easier to combine these sections and reorganize, but I also realize this is a lot of work. Any effort to make the manuscript more accessible will be good.

3. In general it is not a good idea to point to things that are not there in figures. For example, in Figure 3E-D there are arrows and lines showing cells that are not there. I would draw outlines in this case or at least show the merge first. Again, this would make it accessible to a broader audience who are not familiar with AC cell invasion model and what they should look for.

4. It has taken a lot of work for the authors to show that PBAF regulates FOS and BAF cell cycle. Why do they use “?” in the model figure? Maybe use a dotted line? Or may say “other factors?”

5. The discussion is long and can be tightened a bit—this does not mean that they need to remove speculative sections.

**Have all data underlying the figures and results presented in the manuscript been provided?**

Reviewer #1: Yes

Reviewer #2: Yes

Reviewer #3: Yes

PLOS authors have the option to publish the peer review history of their article (what does this mean?). If published, this will include your full peer review and any attached files.

Reviewer #1: No

Reviewer #2: No

Reviewer #3: No

---

## [Editor Report · Decision Letter 1]

7 Dec 2021

Dear Dr Matus,

We are pleased to inform you that your manuscript entitled "The SWI/SNF chromatin remodeling assemblies BAF and PBAF differentially regulate cell cycle exit and cellular invasion in vivo" has been editorially accepted for publication in PLOS Genetics. Congratulations!

Yours sincerely,

Jeremy Nance

Associate Editor

PLOS Genetics

Gregory P. Copenhaver

Editor-in-Chief

PLOS Genetics

Comments from the reviewers (if applicable):

**Data Deposition**

http://datadryad.org/submit?journalID=pgenetics&manu=PGENETICS-D-21-01331R1

**Press Queries**

---

## [Editor Report · Acceptance letter]

22 Dec 2021

PGENETICS-D-21-01331R1 

The SWI/SNF chromatin remodeling assemblies BAF and PBAF differentially regulate cell cycle exit and cellular invasion in vivo 

Dear Dr Matus, 

We are pleased to inform you that your manuscript entitled "The SWI/SNF chromatin remodeling assemblies BAF and PBAF differentially regulate cell cycle exit and cellular invasion in vivo" has been formally accepted for publication in PLOS Genetics! Your manuscript is now with our production department and you will be notified of the publication date in due course.

With kind regards,

Livia Horvath

PLOS Genetics

On behalf of:
